# LLM-JEPA: Large Language Models Meet Joint Embedding Predictive Architectures

**Hai Huang**
Atlassian
hhuang3@atlassian.com

**Yann LeCun**
NYU
yann.lecun@nyu.edu

**Randall Balestriero**
Brown University
rbalestr@brown.edu

## Abstract

Large Language Model (LLM) pretraining, finetuning, and evaluation rely on input-space reconstruction and generative capabilities. Yet, it has been observed in vision that embedding-space training objectives, e.g., with Joint Embedding Predictive Architectures (JEPAs), are far superior to their input-space counterpart. That mismatch in how training is achieved between language and vision opens up a natural question: *can language training methods learn a few tricks from the vision ones?* The lack of JEPA-style LLM is a testimony of the challenge in designing such objectives for language. In this work, we propose a first step in that direction where we develop LLM-JEPA, a JEPA based solution for LLMs applicable both to finetuning and pretraining. Thus far, LLM-JEPA is able to outperform the standard LLM training objectives by a significant margin across models, all while being robust to overfiting. Those findings are observed across numerous datasets (NL-RX, GSM8K, Spider, RottenTomatoes) and various models from the Llama3, OpenELM, Gemma2 and Olmo families. Code: https://github.com/galilai-group/llm-jepa.

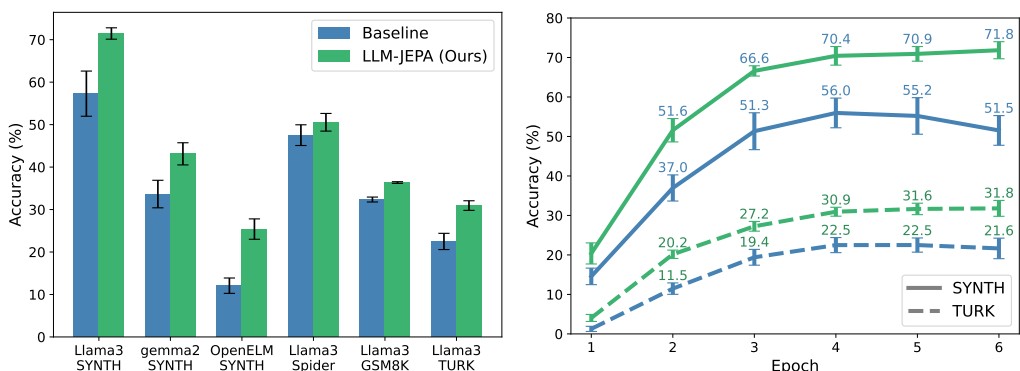

Figure 1: LLM-JEPA produces strong fine-tuned models across datasets and models.

## 1 Introduction

The research landscape around representation learning has been increasingly divided into two camps: (i) generative or reconstruction-based methods (Brown et al., 2020b; Chowdhery et al., 2023; He et al., 2022; LeCun, 2022), and (ii) reconstruction-free Joint Embedding Predictive Architectures (JEPAs) (Assran et al., 2023; Baevski et al., 2022; Bardes et al., 2024). While the former is self-explanatory, the latter learns a representation by ensuring that different *views*, e.g., pictures of a same building at different time of day, can be predicted from each other, all while preventing a collapse of the embeddings. By moving away from input-space objectives, JEPAs training benefits from less biases (Littwin et al., 2024), at the cost of potential dimensional collapse of their representation (Jing et al., 2021; Kenneweg et al., 2025). That divide has been well studied in vision, where it was found that JEPAs offer multiple provable benefits when it comes to knowledge discovery for perception tasks. In the realm of Natural Language Processing however, reconstruction-based methods remain

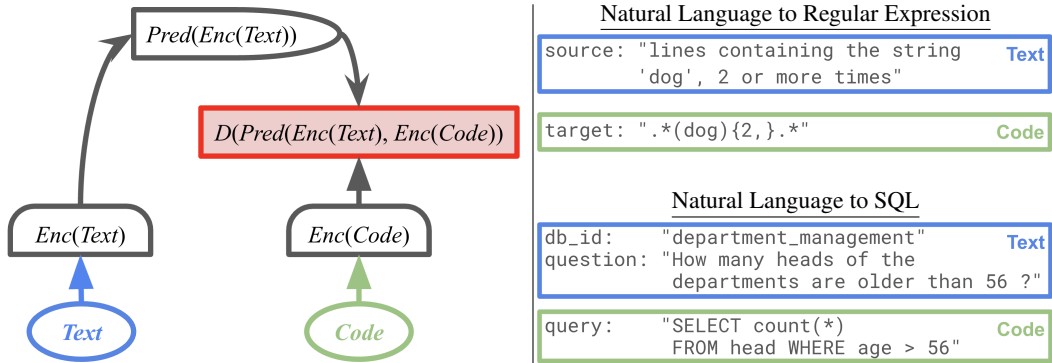

Figure 2: **Left:** JEPA applied to NLP tasks that has $Text$ and $Code$, where $Text$ and $Code$ are naturally two views of the same thing. **Right**: **(top)**: An illustration of the NL-RX-SYNTH dataset, where each sample consists of a description of the regular expression in natural language ($Text$) and the regular expression itself ($Code$). **(bottom)**: The Spider dataset, where $Text$ is the database ID and description of the SQL query and $Code$ is the SQL query itself.

predominant. In fact, today's Large Language Models are mostly judged from their ability to generate samples and answers in input space in text form–making it challenging to leverage JEPA objectives.

Yet, LLMs' task also involve perception and reasoning where JEPA is known to be preferable. It thus seems crucial to adapt JEPA solutions to LLMs in the hope to showcase the same benefits as witnessed in vision. This first step is exactly what we present in this study. We propose to improve the representation quality of LLMs by leveraging a novel objective combining both the original reconstruction based loss–with an additional JEPA objective. To do so, we focus first on tasks and datasets that are inherently suited for JEPA objectives: the ones providing multiple *views* of the same underlying knowledge. One typical example is a git issue and the corresponding code diff (fig. 2) (Jimenez et al., 2024). The two samples are two views–one being plain English and one being in code–of the same underlying functionality. Let's use that particular example to highlight our core contribution:

*Viewing the (text, code) pairs as views of the same underlying knowledge enables JEPA objectives to be utilized with LLMs, complementing the standard text → code generative task.*

We strongly emphasize that being able to obtain non-trivial views, such as described above, is crucial to the success of JEPA objectives. While we restrict ourselves to datasets offering those non-trivial views, developing a mechanism akin to data-augmentation in vision would enable JEPA objectives to be used on any dataset. Nonetheless, we believe that our proposed solution–coined LLM-JEPA–and empirical study will serve as a first step towards more JEPA-centric LLM pretraining and finetuning. We summarize our contributions below:

- **Novel JEPA-based training objective:** We present the first JEPA-based training objective for LLMs operating in embedding space and with different views–perfectly following vision-based JEPAs without sacrificing the generative capabilities of LLMs
- **Improved SOTA:** We empirically validate our formulation in various finetuning settings, where we obtain improvements over standard LLM finetuning solutions. We also explore pretraining scenarios showing encouraging results of LLM-JEPA
- **Extensive empirical validation:** on various model family (llama, gemma, apple/openelm, allenai/olmo), dataset (NL-RX, GSM8K, Spider, RottenTomatoes), and size.

## 2 BACKGROUND

**Large Language Models.** Contemporary LLMs are mostly built from the same core principles: stacking numerous layers of nonlinear operations and skip-connections–known as Transformers. While subtleties may differ, e.g., about positional embeddings, initialization, normalization, the main driver of performance remains the availability of high quality dataset during the pretraining stage. The training objective in itself has also been standardize throughout methods: autoregressive token-space reconstruction. Let's first denote by $\mathcal{L}_{\text{LLM}}$ the typical LLM objective used for the specific task and

dataset at hand. In most cases, this will be a cross-entropy loss between the predicted tokens and the ground-truth token to reconstruction. We note that our LLM-JEPA construction is agnostic of $\mathcal{L}_{\mathrm{LLM}}$ hence making our method general to numerous scenarios.

$$\mathcal{L}_{\mathrm{LLM}}(\mathrm{Text}_{1:L-1}, \mathrm{Text}_L) = \mathrm{CrossEntropy}\left(\mathrm{Classifier}\left(\mathrm{Enc}(\mathrm{Text}_{1:L-1})\right), \mathrm{Text}_L\right), \quad (1)$$

where $\mathrm{Classifier}(\cdot)$ predicts the logits of the next token $\mathrm{Text}_L$ given the past tokens $\mathrm{Text}_{1:L-1}$. Computation of eq. (1) is done at once over $L$ through causal autoregression. Different stages and tasks may vary the input and output of the loss.

**Alternative training objectives.** Next token prediction (NTP) is the prevalent pretraining solution for today's latest LLMs. There exists a few alternatives, e.g., SimCSE leverages a contrastive loss in the latent space by treating different dropout-induced views of the same sentence as positive pairs, resulting in state-of-the-art sentence embeddings quality (Gao et al., 2021). In a similar spirit, (Wang et al., 2022) uses weak supervision from text pairs to learn a joint embedding architecture. An alternative relying on pretrained models instead of the supervised text pairs was explore in (Ni et al., 2021). While those approaches are powerful, they are all concern with producing text embeddings without generative capabilities which greatly limits the applicability of those models since numerous evaluations and use-cases require generation–which is core to our proposed method. Another solution employs BERT pretraining coupled with a latent space semantic loss to ensure that representations of semantically similar sentences are nearby in embedding space. This additional term led to improved performance on semantic tasks compared to masked language modeling alone (Reimers & Gurevych, 2019)–yet again by building atop BERT this solution prevents generative evaluation and use.

## 3 JEPA-LLM: IMPROVING LLMS' REASONING AND GENERATIVE CAPABILITIES

We propose the LLM-JEPA loss in section 3.1 along with extensive empirical validations in section 4 demonstrating clear finetuning and pretraining benefits.

### 3.1 THE LLM-JEPA OBJECTIVE

Throughout this section, we will use Text and Code as concrete examples of having different views of the same underlying knowledge. It should be clear to the reader that our proposed LLM-JEPA objective handles different types of views similarly.

The construction of our LLM-JEPA objective relies on two principles. First, we must preserve the generative capabilities of LLMs and we therefore start with the $\mathcal{L}_{\mathrm{LLM}}$ from eq. (1). Second, we aim to improve the abstraction capabilities of LLMs using the joint embedding prediction task. On top of $\mathcal{L}_{\mathrm{LLM}}$, we then propose to add the well-established JEPA objective leading to the complete loss $\mathcal{L}$ defined as

$$\mathcal{L}_{\mathrm{LLM-JEPA}} = \underbrace{\sum_{\ell=2}^{L} \mathcal{L}_{\mathrm{LLM}}(\mathrm{Text}_{1:\ell-1}, \mathrm{Text}_\ell)}_{\text{generative capabilities (LLM)}} + \lambda \times \underbrace{d(\mathrm{Pred}(\mathrm{Enc}(\mathrm{Text})), \mathrm{Enc}(\mathrm{Code}))}_{\text{abstraction capabilities (JEPA)}}, \quad (2)$$

where $\lambda \geq 0$ is an hyperparameter balancing the contribution of the two terms, Pred and Enc are the predictor and encoder networks respectively, and $d$ is a metric of choice, e.g., the $\ell_2$ distance. Let's now precisely describe each of those components.

**The encoder.** We use the `hidden_state` of the last token from the last layer as the embedding of an input sequence–as commonly done for LLM probing. We pack both Text and Code into a single context window, applying an attention mask to ensure they do not reference each other. Implementation-wise, most HuggingFace `transformers` support an additive attention mask, where setting entry $(i, j) = -\infty$ prevents token $j$ from attending to token $i$ (for $i < j$). Using this mechanism, we implement LLM-JEPA with only one additional forward pass. This introduces extra cost during training, but not during inference—see section 6 for further discussion.

**The metric.** When it comes comparing embeddings, it is now widely accepted in vision to leverage the cosine similarity. We thus propose to do the same for LLM-JEPA.

**The predictor.** We leverage the auto-regressive nature of LLM and their internal self-attention to define a *tied-weights predictor*. By introducing a special token [PRED] at the end of a given Text, we allow for further nonlinear processing of the input hereby producing $Pred(\cdot)$ at the final embedding (the hidden state of the last [PRED] token) of the last layer. By reusing the internal weights of the LLM for the prediction task, we greatly reduce the training overhead and architectural design choices. Practically, we append $k \in \{0, \dots, K\}$ *predictor tokens* to an input prompt and use the embedding of the last predictor token to be $Pred(Enc(\cdot))$. When $k = 0$, the predictor simply becomes an identity mapping, i.e., $Pred(x) = x$.

### 3.2 IMPLEMENTATION WITH CUSTOM ATTENTION MASK

The most important challenge in implementing our proposed LLM-JEPA objective lies in obtaining the embeddings of the different views, e.g., Text and Code in eq. (2). A priori, it is not possible to get them in one forward pass because the current self-attention–albeit being causal. Even if we were to concatenate the two views which is already done for the next token prediction part of the loss, it would make the representation of the second view rely on the first view. As a result, we propose the following custom self-attention mask that is causal per block, with number of blocks set to 2:

```python
def additive_mask(k: int):
    """Returns a k by k triangle mask."""
    mask = torch.zeros((k, k), dtype=torch.float32)
    mask[torch.triu(torch.ones(k, k), diagonal=1) == 1] = -torch.inf
    return mask

# Initialize all elemets to -inf.
mask = torch.full((batch_size * 2, 1, seq_length, seq_length), -torch.inf
    ).to(device)
# Text starts from t_start and is of size t_size, and Code starts
# from c_start and is of size c_size. Set for the i-th batch.
mask[i, :, t_start: t_start + t_size, t_start: t_start + t_size] =
    additive_mask(t_size)
mask[i, :, c_start: c_start + c_size, c_start: c_start + c_size] =
    additive_mask(c_size)
```

By leveraging the above implementation, we are able to obtain the LLM-JEPA loss value in one extra forward passes instead of two. While this is still a substantial slowdown, we will explore later in the manuscript a *dropout* version where the JEPA term is only applied some % of the mini batch–the end result will be that are comparable FLOPS the proposed LLM-JEPA is still able to outperform the baseline.

**Relation to Previous Work.** Because loss functions such as $\mathcal{L}_{\text{LLM}}$ (input space reconstruction since tokens are lossless compression of the original prompts) have been shown to be sub-optimal in vision, a few LLM variations have started to employ embedding space regularizers and training objectives (Barrault et al., 2024; Wang & Sun, 2025). Current solution however rely on intricate structural constraints of the embedding space, e.g., hierarchical organization and cluster, and thus fall out of the JEPA scope. We also note that our interpretation of *views* when it comes to LLM datasets, e.g., (text issue, code diff), is something that has been leveraged as part of the LLM finetuning solutions–by learning to generate one from the other–without a JEPA-style loss. This includes natural language to regular expression translation (Locascio et al., 2016; Ye et al., 2020; Zhong et al., 2018), natural language to SQL parsing (Guo et al., 2019; Iyer et al., 2017; Li et al., 2023; Wang et al., 2019; Yu et al., 2018) and the more recent issue descriptions to code diffs (Cabrera Lozoya et al., 2021; Hoang et al., 2020; Tian et al., 2020; Zhou et al., 2023). More intricate examples involve text-based problem solving and their counterpart program induction (Amini et al., 2019; Cobbe et al., 2021; Hendrycks et al., 2021; Ling et al., 2017).

### 3.3 A GOOD NEXT TOKEN PREDICTOR IS NOT A GOOD JEPA

Before validating the proposed LLM-JEPA to real task and models, we ask ourselves a simple question. Is it really necessary to have an additional JEPA term? Is that term already implicitly minimized by the original next token prediction objective?

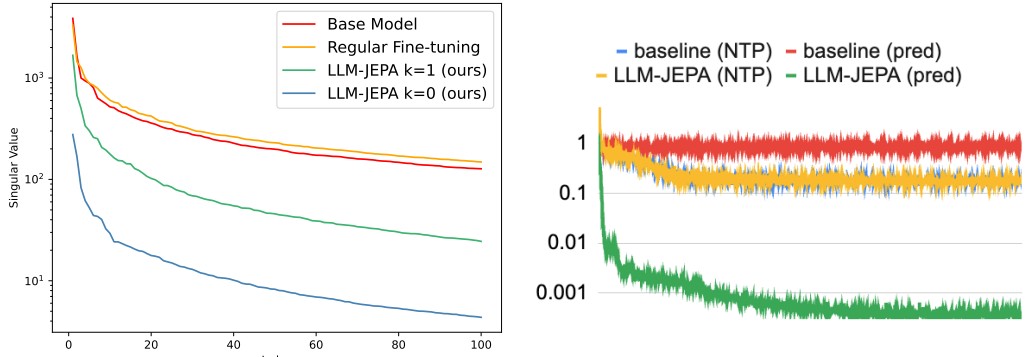

Figure 3: **left:** The top 100 singular values of $\mathrm{Enc}(\mathrm{Text}) - \mathrm{Enc}(\mathrm{Code})$. The curves of LLM-JEPA (ours) are a few magnitudes lower than that of base model and regular fine-tuning, meaning the mapping from $\mathrm{Text}$ to $\mathrm{Code}$ are confined within a narrow subspace, fostering the nice structure we see in Figure 4. **Right:** Losses in fine-tuning with $\mathcal{L}_{\mathrm{LLM}}$ loss ($\mathcal{L}_{\mathrm{LLM}}$) and $\mathcal{L}_{\mathrm{LLM-JEPA}}$ loss ($\mathcal{L}_{\mathrm{LLM-JEPA}}$, our method). We measure both the cross-entropy loss for next token prediction ($Loss_{LLM}$, $\mathcal{L}_{\mathrm{LLM}}$ in chart) and JEPA prediction loss ($D(\cdot, \cdot)$, **pred** in chart), although the latter does not contribute in the baseline case. The accuracy is $51.95\%$ for $\mathcal{L}_{\mathrm{LLM}}$ and $71.10\%$ for $\mathcal{L}_{\mathrm{LLM-JEPA}}$. Since $\mathcal{L}_{\mathrm{LLM}}$ and $\mathcal{L}_{\mathrm{LLM-JEPA}}$ share similar $\mathcal{L}_{\mathrm{LLM}}$ loss, the $\mathcal{L}_{\mathrm{LLM}}$ loss cannot explain the gap between the accuracy. **pred** stays a constant in $\mathcal{L}_{\mathrm{LLM}}$, while is minimized in $\mathcal{L}_{\mathrm{LLM-JEPA}}$, hence **pred** should be the main reason behind the accuracy gap.

To answer that, we compare two controlled settings. We will be using Llama-3.2-1B-Instruct and NL-RX-SYNTH and have two training setup. In the first, we do the usual next-token prediction loss but monitor the JEPA objective, i.e., no gradient comes from the JEPA loss. In the second, we will use the proposed LLM-JEPA for gradient computation. We also monitor the prediction loss in both cases. We obtain the following finding in fig. 3: minimizing $\mathcal{L}_{\mathrm{LLM}}$ *does not* implicitly minimize $\mathcal{L}_{\mathrm{JEPA}}$–indicating that it is required to add that term during training. This can be seen by comparing the red and green lines. A natural follow-up question is *are we trading off next token prediction for JEPA?*. In the same fig. 3 we obtain that the next token prediction capability is not hindered by the presence of the JEPA term (compare the blue and yellow lines that are overlapping). This is a very important observation echoing what was empirically observed in (Balestriero & LeCun, 2024) where it was observed that autoencoders in image space could become much stronger classifiers without sacrificing the reconstruction and generative objective. All in all we obtain that employed LLM-JEPA only brings additional structure of the LLM latent space without altering its generative capabilities. As we will see in the follows sections, we indeed obtain much better performances that the baseline–in the generative evaluation setup.

## 4 EMPIRICAL VALIDATION: LLM-JEPAS OUTPERFORM LLMS

In this section, we first validate the proposed LLM-JEPA across four model families and four datasets with natural two-view structures (section 4.1). The consistent improvements observed across the board motivate us to further examine the internal representations of LLM-JEPA to better understand the source of its strength (section 4.2). We conclude this section with a rigorous ablation study on key design choices (section 4.3).

We adopt a universal protocol across all experiments by using five fixed random seeds, $\{82, 23, 37, 84, 4\}$, for training. Each experiment is repeated five times, once with each seed. This setup enables us to assess the stability of LLM-JEPA and to conduct paired one-tailed $t$-tests for statistical significance.

### 4.1 FINE-TUNING AND PRETRAINING STRONGER GENERATIVE MODELS VIA JEPA

**LLM-JEPA Improves Finetuning.** We run experiments across multiple pretrained LLMs (Llama-3.2-1B-Instruct (Grattafiori et al., 2024), gemma-2-2b-it (Team et al., 2024), OpenELM-1_1B-Instruct (Mehta et al., 2024), and OLMo-2-0425-1B-Instruct (OLMo et al., 2024)) with various datasets

(NL-RX-SYNTH, NL-RX-TURK (Locascio et al., 2016), GSM8K (Cobbe et al., 2021), Spider (Yu et al., 2018)).

To demonstrate that LLM-JEPA improves from the strongest possible baseline, we first search for the best learning rate $lr \in \{1e-5, 2e-5, 4e-5, 8e-5\}$ by selecting the value that yields the highest accuracy of $\mathcal{L}_{\mathrm{LLM}}$ after 4 epochs for a given (model,dataset) pair. Then we tune the hyperparameter specific to $\mathcal{L}_{\mathrm{LLM-JEPA}}$, $k$ and $\lambda$ in a two dimensional grid defined by $(k, \lambda) \in \{0, 1, 2, 3, 4\} \times \{0.5, 1, 2, 4\}$ (fig. 7 and table 12 in the Appendix, also appendix A.8 presents an empirical approach to efficiently identify the optimal $(\lambda, k)$). For both NL-RX-SYNTH and NL-RX-TURK, accuracy is exact match of the generated regular expression; for GSM8K, accuracy is exact match of the final result; and for Spider, accuracy is exact match of the execution result of the generated query.

We provide results demonstrating that LLM-JEPA improves performances across

- four model families, see fig. 1 left and table 14 in the Appendix.
- four datasets, see table 15 in the Appendix.
- 6 training epochs (fig. 1 right). We also observe that LLM-JEPA resists overfitting, whereas standard fine-tuning does not.
- four different sizes: 1B, 3B, 7B, and 8B, see table 17 in the Appendix

Examples of inputs, targets, model predictions, and error analyses are provided in table 1 (more examples can be found in table 13 in the Appendix). **For LoRA fine-tuning**, the performance gains of LLM-JEPA hold consistently across different LoRA ranks (table 10 in the Appendix). We also observe the same resistance to overfitting in the LoRA setting (fig. 8 in the Appendix).

Table 1: Regular expressions generated by Llama-3.2-1B-Instruct after fine-tuning with $\mathcal{L}_{\mathrm{LLM}}$ loss and $\mathcal{L}_{\mathrm{LLM-JEPA}}$ loss (ours). Color code: wrong , extra , missing

| Ground Truth | $\mathcal{L}_{\mathrm{LLM}}$ | $\mathcal{L}_{\mathrm{LLM-JEPA}}$ (ours) |
|---|---|---|
| **lines not having the string 'dog' followed by a number, 3 or more times** | | |
| ((dog.*[0-9].*){3,}) | ((dog.*[0-9].*){3,}) | ((dog.*[0-9].*){3,}) |
| **lines containing ending with a vowel, zero or more times** | | |
| .*(.*)(([AEIOUaeiou])*).* | (.*)(([AEIOUaeiou])*) * | ( .* ) (.*) { ([AEIOUaeiou])* } |
| **lines with a number or a character before a vowel** | | |
| ((([0-9])\|(.)).*[AEIOUaeiou]).* | (([0-9])\|(.)).*([AEIOUaeiou]).* .* | ((([0-9])\|(.)).*([AEIOUaeiou]).* |
| **lines with words with the string 'dog', a letter, and a number** | | |
| (((([0-9])&(dog))\|([A-Za-z]))* | (((([0-9])&(dog))\|([A-Za-z]))* | (( [0-9])&(dog))\|( ( [A-Za-z]) * ) |

**LLM-JEPA Improves Pretraining.** A natural extension of our fine-tuning results is to examine pretraining. We pretrain Llama-3.2-1B-Instruct from randomly initialized weights on the NL-RX-SYNTH dataset. Owing to the limited size of the dataset, the pretrained model fails to reliably learn how to terminate generation. To address this, we adjust the evaluation criterion, deeming a generated solution valid as long as it begins with the ground-truth sequence. We obtain that LLM-JEPA also improves the quality of the learned representation, as shown in table 2.

Table 2: Pretraining accuracy on dataset NL-RX-SYNTH by Next Token Prediction ($\mathcal{L}_{\mathrm{LLM}}$) loss vs. $\mathcal{L}_{\mathrm{LLM-JEPA}}$ loss (our method). We inherit the best configuration from fine-tuning. Each case runs five times. Average accuracy and standard deviation are reported. We also report $p$-value of paired, single-tailed $t$-Test.

| Model | Method | Accuracy (%) ↑ | $p$-value ↓ | Config |
|---|---|---|---|---|
| Llama-3.2-1B-Instruct | $\mathcal{L}_{\mathrm{LLM}}$ | $54.38 \pm 1.70$ | $2.94e-4$ | $lr = 8e-5$ |
| | $\mathcal{L}_{\mathrm{LLM-JEPA}}$ (ours) | $60.59 \pm 1.01$ | | $\lambda = 2, k = 3$, same $lr$ |

**We then conduct a more advanced pretraining experiment** on dataset cestwc/paraphrase containing groups of 5 paraphrases. We leverage the five paraphrases to construct the JEPA loss by having the

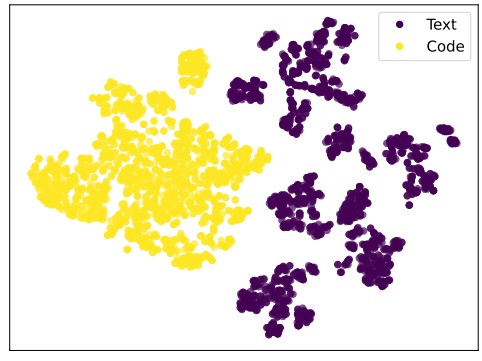 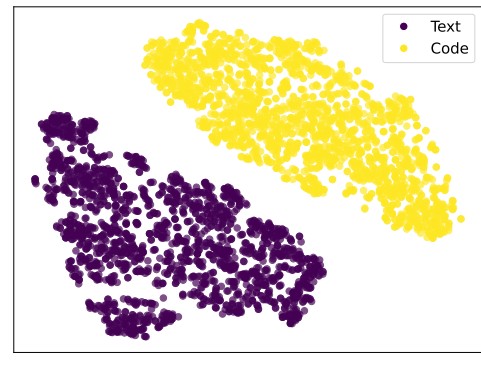

(a) Baseline: Fine-tuned by NTP loss    (b) LLM-JEPA (Ours) $k = 0$

Figure 4: $t$-SNE plot of Text and Code representations in **(a)** Baseline that is fine-tuned with Next Token Prediction (NTP) loss, **(b)** LLM-JEPA (ours) with $k = 0$. Clearly LLM-JEPA (ours) induced nice structure on the representations while fine-tuning with NTP loss disrupted the structure in the base model. A full version of this figure is in appendix A.2.

$i$-th version of a paraphrase predict the $(i + 1)$-th version. The goal is to encourage the JEPA loss to tie the embeddings of all versions into a compact subspace, providing a geometric foundation to align their representations. We pretrain the model for four epochs and then evaluate it on Rotten Tomatoes and Yelp after one epoch of fine-tuning. Although there is no direct link between the pretraining and evaluation datasets, we show that LLM-JEPA pretraining yields statistically significant improvements in downstream performance (see table 11 in the Appendix). Note that fine-tuning does not employ the JEPA loss—highlighting that the benefits arise specifically from the pretraining stage.

For Rotten Tomatoes and Yelp, we fine-tune the model to generate discrete sentiment labels: `Good` and `Bad` for Rotten Tomatoes, and `Very Good`, `Good`, `Mediocre`, `Bad`, and `Very Bad` for Yelp. A prediction is deemed correct if the generated output begins with the ground-truth label. This evaluation approach—mapping free-form generation to categorical labels and applying prefix matching—follows common practice in prior work on text classification with generative models (McCann et al., 2018; Raffel et al., 2020; Wei et al., 2022).

Lastly, we provide in table 9 (in the Appendix) generated samples demonstrating that JEPA pretraining does maintain the generative capabilities of the model when prompted with the first few tokens in the cestwc/paraphrase dataset.

## 4.2 STRUCTURED REPRESENTATIONS INDUCED BY LLM-JEPA

We also examine the representation space to better understand how LLM-JEPA regularizes learned features. Specifically, we plot $t$-SNE embeddings for both Text and Code across three settings: the base model, a model fine-tuned with $\mathcal{L}_{\text{LLM}}$, and a model fine-tuned with $\mathcal{L}_{\text{LLM-JEPA}}$. As shown in fig. 4, clear structure emerges after fine-tuning with $\mathcal{L}_{\text{LLM-JEPA}}$. We hypothesize that $\mathcal{L}_{\text{LLM-JEPA}}$ enforces structure in the representation space by constraining the mapping from $\text{Enc}(\text{Text})$ to $\text{Enc}(\text{Code})$ within a narrow subspace. If this is the case, the SVD decomposition of $\text{Enc}(\text{Text}) - \text{Enc}(\text{Code})$ should yield significantly smaller singular values, which is confirmed in fig. 3. Furthermore, we hypothesize that the mapping is approximately linear. To test this, we compute the least-squares regression error, and table 16 in the Appendix supports this hypothesis. Together, these results suggest that LLM-JEPA promotes a near-linear transformation between Text and Code representations, which may underlie its accuracy improvements.

## 4.3 ABLATION STUDY ON DESIGN CHOICES

In this section, we examine several alternative design choices. Specifically, we compare the use of cosine similarity against $\ell_2$-norm and mean squared error (MSE); appending a [PRED] token to the end of Text versus prepending it; and using Text to predict Code versus the reverse direction

(Code → Text). As shown in table 3, none of these alternatives perform as well as LLM-JEPA, although some outperform standard fine-tuning.

Table 3: Fine-tuning accuracy on NL-RX-SYNTH under different design choices. Reported are average accuracy and standard deviation across runs. Learning rate $lr = 2e-5$, $\lambda = 1.0$, and $k = 1$. The 2.22 for $\ell_2$-norm is not a typo. See appendix A.11 for discussion.

| | | | Accuracy (%) ↑ | | | |
|---|---|---|---|---|---|---|
| Baseline | LLM-JEPA | $\ell_2$-norm | MSE | Prepend | Code → Text | InfoNCE loss |
| $57.29 \pm 5.32$ | $71.46 \pm 1.34$ | $2.22 \pm 0.07$ | $70.64 \pm 2.05$ | $68.07 \pm 2.57$ | $65.70 \pm 2.63$ | $34.40 \pm 6.10$ |

Additionally, we replace our cosine similarity loss with the InfoNCE loss (van den Oord et al., 2018), which results in lower accuracy compared to the baseline. Moreover, its standard deviation is substantially higher than that of other alternatives. We use the commonly adopted temperature of $\tau = 0.07$ for the InfoNCE loss (Chen et al., 2020; Radford et al., 2021). We performed an additional ablation in which we replaced the hidden state of the last token with the average of all hidden states over all tokens. This modification degraded performance to $65.46 \pm 3.51$.

We hypothesize that the accuracy improvements stem from **representation alignment**—i.e., the model learns to collapse representations of semantically similar text into a tightly aligned region (often nearly a line), which facilitates extrapolation and improves generalization. Under this view, InfoNCE can be detrimental because its contrastive objective explicitly pushes representations apart, counteracting the alignment needed for extrapolative behavior. We validated this hypothesis through additional experiments; please see appendix A.10 for details.

For $k > 0$, we append multiple predictor tokens: [PRED_1], ..., [PRED_k], to the text and use the hidden state of [PRED_k] as the predicted representation. To better understand the mechanism, we examined whether performance gains come primarily from (i) the **number** of predictor tokens or (ii) the **variety** of distinct predictor tokens. We conducted ablations using identical predictor tokens, and the results (table 4) show only minor differences. This suggests that the dominant factor is simply increasing the number of prediction steps (FLOPs), rather than requiring distinct token embeddings.

Table 4: Comparison of distinct vs. identical [PRED] tokens. Both settings achieve comparable performance.

| Config | Distinct [PRED] | Identical [PRED] |
|---|---|---|
| **Llama 3.2 / GSM8K** | $36.36 \pm 0.20$ | $36.74 \pm 0.70$ |
| **OpenELM / SYNTH** | $25.40 \pm 2.40$ | $25.01 \pm 1.60$ |

An alternative design is to replace the [PRED] tokens with a standalone linear predictor network. A linear predictor is sufficient because the mapping from Text to Code representations is nearly linear (table 15 and fig. 3 left). However, in the fine-tuning regime this approach suffers from a **cold-start problem**—the linear head must be trained entirely from scratch. Empirically, a straightforward linear predictor yields consistently subpar performance compared to using [PRED] tokens (table 5).

Table 5: Trainable linear predictor yields consistently subpar performance compared to using [PRED]tokens.

| Linear $\lambda = 0.5$ | Linear $\lambda = 1$ | Linear $\lambda = 2$ | [PRED] $\lambda = 1, k = 1$ |
|---|---|---|---|
| $70.16 \pm 1.87$ | $67.93 \pm 2.70$ | $64.54 \pm 4.99$ | $71.46 \pm 1.34$ |

## 5 TOWARDS FASTER AND MORE GENERAL LLM-JEPAS

The structured representations induced by LLM-JEPA have the potential to provide universal benefits across diverse LLM applications. In this section, we explore its limits by evaluating datasets without natural two-view structures and models with richer capabilities (section 5.1). The promising results further motivate us to investigate methods for accelerating LLM-JEPA (section 5.2), as computational overhead remains a key obstacle to broad adoption.

Table 6: Fine-tuning accuracy of Llama-3.2-1B-Instruct with Next Token Prediction ($\mathcal{L}_{\text{LLM}}$) loss vs. $\mathcal{L}_{\text{LLM}-\text{JEPA}}$ loss (our method). Each case runs five times. Average accuracy and standard deviation are reported. We also report $p$-value of paired, single-tailed $t$-Test.

| Dataset | Method | Accuracy (%) ↑ | $p$-value ↓ | Config |
|---------|--------|----------------|-------------|--------|
| NQ-Open | $\mathcal{L}_{\text{LLM}}$
$\mathcal{L}_{\text{LLM}-\text{JEPA}}$ (ours) | $20.12 \pm 0.41$
$21.59 \pm 0.40$ | $2.44e-3$ | $lr = 2e-5$
$\lambda = 1024, k = 0$, same $lr$ |
| HellaSwag | $\mathcal{L}_{\text{LLM}}$
$\mathcal{L}_{\text{LLM}-\text{JEPA}}$ (ours) | $27.93 \pm 0.46$
$35.22 \pm 2.09$ | $8.12e-4$ | $lr = 4e-5$
$\lambda = 1, k = 4$, same $lr$ |

Table 7: Fine-tuning accuracy of GSM8K with Next Token Prediction ($\mathcal{L}_{\text{LLM}}$) loss vs. $\mathcal{L}_{\text{LLM}-\text{JEPA}}$ loss (our method). Each case runs five times. Average accuracy and standard deviation are reported. We also report $p$-value of paired, single-tailed $t$-Test.

| Model | Method | Accuracy (%) ↑ | $p$-value ↓ | Config |
|-------|--------|----------------|-------------|--------|
| Qwen3-1.7B | $\mathcal{L}_{\text{LLM}}$
$\mathcal{L}_{\text{LLM}-\text{JEPA}}$ (ours) | $44.32 \pm 0.39$
$45.00 \pm 0.40$ | $0.0115$ | $lr = 4e-5$
$\lambda = 1, k = 0$, same $lr$ |
| R1-Distill-Qwen-1.5B | $\mathcal{L}_{\text{LLM}}$
$\mathcal{L}_{\text{LLM}-\text{JEPA}}$ (ours) | $13.87 \pm 1.01$
$15.04 \pm 0.15$ | $0.0396$ | $lr = 8e-5$
$\lambda = 0.5, k = 1$, same $lr$ |

## 5.1 Beyond Code: Applying LLM-JEPA to Q&A Datasets and Reasoning Models

**We evaluate Llama-3.2-1B-Instruct on two QA benchmarks**: NQ-Open (Lee et al., 2019a) and HellaSwag (Zellers et al., 2019). Our results show that LLM-JEPA achieves statistically significant improvements on both datasets, demonstrating its capability beyond the canonical setup where Text and Code are treated as two complementary views of the same object.

For NQ-Open, we regard Text as the question and Code as the answer span, typically consisting of only a few tokens, which contrasts with the more balanced sequence lengths found in other datasets. HellaSwag, by contrast, is a context-completion multiple-choice task. Rather than defining Code as the answer label (a single token from $\text{A}, \text{B}, \text{C}, \text{D}$), we instead let Text denote the context and Code represent the correct continuation. This formulation differs from prior setups in two important ways: (i) Both Text and Code are now integral components of the question, and (ii) The relationship between context and completion is more diverse than the near-equivalence seen in NL→Regex or NL→SQL mappings.

Despite these differences, LLM-JEPA consistently improves accuracy on both benchmarks table 6.

For **NQ-Open**, generated answers may include additional syntactic or supporting tokens. Following prior work, we deem a prediction correct if *any ground-truth answer appears as a substring* of the generated output (Lee et al., 2019b; Guu et al., 2020; Izacard & Grave, 2021). For **HellaSwag**, we compute the relative probabilities of the four candidate options $\text{A}, \text{B}, \text{C}, \text{D}$ and select the answer with the highest probability, consistent with standard practice in multiple-choice language understanding benchmarks (Zellers et al., 2019; Brown et al., 2020a; OpenAI, 2023).

An additional observation is that performance continues to improve as we scale $\lambda$ up to 1024, without encountering a plateau. While table 12 in the Appendix suggests that extreme values of $\lambda$ can degrade accuracy, in certain cases it remains beneficial to push $\lambda$ further, yielding extra gains.

**We then evaluate two strong reasoning models**—Qwen3-1.7B (Yang et al., 2025) and DeepSeek-R1-Distill-Qwen-1.5B (DeepSeek-AI et al., 2025)—on GSM8K. As shown in table 7, both models achieve statistically significant improvements when augmented with LLM-JEPA. These results offer promising evidence that LLM-JEPA extends its benefits to large reasoning models (LRMs).

## 5.2 Faster LLM-JEPAs via Loss Dropout

To further reduce compute, we introduce **random JEPA-loss dropout** ($LD$). During training or fine-tuning, we randomly drop the JEPA loss at a specified $LD$ rate. Loss dropout is applied at the

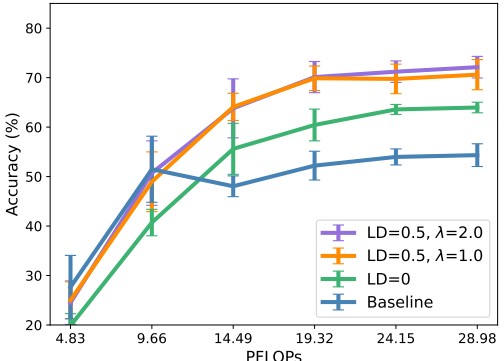

Figure 5: LLM-JEPA converges faster than regular fine-tuning at the same PFLOPs. Furthermore, random JEPA-loss dropout (LD) helps save PFLOPs and boost accuracy at the same amount of compute. $LD = 0$ is the regular LLM-JEPA. Learning rate $lr = 2\mathrm{e}{-5}$ and $k = 1$. $\lambda$ varies.

Table 8: Random JEPA-loss dropout ($LD$) help save PFLOPs and at the same time, boost accuracy. $LD = 0$ is the regular LLM-JEPA. Reported are average accuracy and standard deviation across runs. Each row is 4.83 PFLOPs apart. Learning rate $lr = 2\mathrm{e}{-5}$ and $k = 1$. $\lambda$ varies.

| | | Accuracy (%) ↑ | | | |
| $LD$=0, $\lambda$=1 | $LD$=0.5, $\lambda$=1 | $LD$=0.5, $\lambda$=2 | $LD$=0.75, $\lambda$=1 | $LD$=0.75, $\lambda$=2 | $LD$=0.75, $\lambda$=4 |
| --- | --- | --- | --- | --- | --- |
| $19.85 \pm 2.44$ | $25.00 \pm 3.73$ | $24.50 \pm 4.40$ | $32.46 \pm 3.32$ | $32.10 \pm 3.11$ | $31.45 \pm 3.34$ |
| $40.70 \pm 2.67$ | $48.96 \pm 6.03$ | $50.71 \pm 6.52$ | $53.77 \pm 5.53$ | $57.03 \pm 3.51$ | $56.75 \pm 4.63$ |
| $55.60 \pm 5.16$ | $64.08 \pm 2.75$ | $63.79 \pm 5.96$ | $64.51 \pm 7.28$ | $67.03 \pm 3.08$ | $65.32 \pm 3.54$ |
| $60.43 \pm 3.21$ | $69.87 \pm 2.48$ | $70.11 \pm 3.15$ | $67.80 \pm 4.94$ | $66.80 \pm 4.06$ | $68.93 \pm 4.59$ |
| $63.57 \pm 1.01$ | $69.74 \pm 2.99$ | $71.20 \pm 2.17$ | $70.00 \pm 4.74$ | $70.77 \pm 4.08$ | $72.42 \pm 1.28$ |
| $63.96 \pm 1.07$ | $70.60 \pm 3.05$ | $72.11 \pm 2.18$ | $70.31 \pm 4.64$ | $70.92 \pm 4.62$ | $73.08 \pm 1.28$ |

batch level. When active, it eliminates the need for an extra forward pass to compute $\mathrm{Enc}(\mathrm{Text})$ and $\mathrm{Enc}(\mathrm{Code})$, thereby saving compute. If $LD = \alpha$, the per-epoch cost becomes $2 - \alpha$ times that of standard fine-tuning, since each batch saves $\alpha$ forward passes. As shown in Figure 5 and Table 8, LLM-JEPA tolerates aggressive loss dropout rates (e.g., 0.5 or 0.75), which leads to higher accuracy under the same compute budget. Moreover, increasing $\lambda$ in proportion to the dropout rate can further improve performance. Empirically, we observe that keeping $\lambda \times (1 - \alpha)$ approximately constant provides a useful guideline for co-tuning $\lambda$ and $\alpha$ to balance compute efficiency and accuracy. The use of the loss dropout coupled with our custom attention mask offers some positive perspectives to further scale LLM-JEPA to full scale pretraining with minimal computational overhead. See appendix A.9 for additional experiments at extremely high dropout $1 - 0.0625$.

## 6 CONCLUSION AND FUTURE WORK

We introduced an alternative training objective for LLMs leveraging JEPAs. Our formulation is an exact replicate of the JEPA objective extensively used in vision–but that hadn't been adapted to language yet. Crucially, our proposed LLM-JEPA maintains the generative capabilities of LLMs while improving their abstract prompt representation as empirically validated across datasets and models. While our experiments mostly focus on finetuning, preliminary pretraining experiment are promising which we plan to scale and more thoroughly test in future work. Regarding the limitations of LLM-JEPA, the primary bottleneck at present is the 2-fold increase in compute cost during training, which is mitigated by random loss dropout.

**Limitations** Despite its strong accuracy gains, LLM-JEPA introduces two additional hyperparameters. As shown in fig. 7, the optimal configuration may occur at any point in a grid $(\lambda, k)$, which imposes a significant cost for hyperparameter tuning. While we have not identified an efficient method to explore this space, we empirically observe that adjacent grid points often yield similar accuracy, suggesting the potential for a more efficient tuning algorithm.

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

Table 9: Generated samples by model pretrained by cestwc/paraphrase dataset. The pretrained model is not good at terminating sentence. prompt and generation

|  | Ground Truth vs. Generation |
| --- | --- |
| Ground Truth | A garden of flowers and a bench stating "City of London." |
| Generation | A garden of flowers and a vase with a flower in it............. |
| Ground Truth | A person that is riding on a horse in a grass field. |
| Generation | A person that is riding in a field................. |
| Ground Truth | A man is riding a horse in a field. |
| Generation | A man is riding a horse in a field................ |
| Ground Truth | There are two birds standing on top of a building |
| Generation | There are two birds standing on a rock................. |
| Ground Truth | Two hawks sit on top of a roof spire. |
| Generation | Two hawks sit on top of a wooden bench................ |
| Ground Truth | .A young woman serving herself at a cookout. |
| Generation | .A young woman serving herself in a kitchen................. |
| Ground Truth | 2 bowls of fruit sit on a table. |
| Generation | 2 bowls of fruit sit on a table................. |
| Ground Truth | A wooden bench written 'CITY OF LONDON' at the park |
| Generation | A wooden bench written 'CITY and a tree................. |

Table 10: Fine-tuning accuracy on dataset NL-RX-SYNTH, LoRA vs. full fine-tuning, both by $\mathcal{L}_{\mathrm{LLM}}$ loss and $\mathcal{L}_{\mathrm{LLM-JEPA}}$ loss (our method). Configuration is $lr = 2e-5, \lambda = 1, k = 1$. Each cell runs five times. Average accuracy and standard deviation are reported. At every LoRA rank, $\mathcal{L}_{\mathrm{LLM-JEPA}}$ (ours) has better accuracy. At LoRA rank 512 (22.59% trainable parameters), $\mathcal{L}_{\mathrm{LLM-JEPA}}$ (ours) achieves same accuracy as full fine-tuning, but $\mathcal{L}_{\mathrm{LLM}}$ still has a significant gap from full fine-tuning.

| LoRA Rank | Method | Accuracy (%) ↑ |
| --- | --- | --- |
| 32 | $\mathcal{L}_{\mathrm{LLM}}$ | $6.09 \pm 0.55$ |
|  | $\mathcal{L}_{\mathrm{LLM-JEPA}}$ (ours) | $7.45 \pm 1.87$ |
| 64 | $\mathcal{L}_{\mathrm{LLM}}$ | $21.09 \pm 1.90$ |
|  | $\mathcal{L}_{\mathrm{LLM-JEPA}}$ (ours) | $32.46 \pm 1.26$ |
| 128 | $\mathcal{L}_{\mathrm{LLM}}$ | $34.21 \pm 2.82$ |
|  | $\mathcal{L}_{\mathrm{LLM-JEPA}}$ (ours) | $48.45 \pm 3.66$ |
| 256 | $\mathcal{L}_{\mathrm{LLM}}$ | $45.57 \pm 4.52$ |
|  | $\mathcal{L}_{\mathrm{LLM-JEPA}}$ (ours) | $60.80 \pm 2.31$ |
| 512 | $\mathcal{L}_{\mathrm{LLM}}$ | $50.18 \pm 5.15$ |
|  | $\mathcal{L}_{\mathrm{LLM-JEPA}}$ (ours) | $72.41 \pm 2.94$ |
| Full | $\mathcal{L}_{\mathrm{LLM}}$ | $57.29 \pm 5.32$ |
|  | $\mathcal{L}_{\mathrm{LLM-JEPA}}$ (ours) | $70.42 \pm 2.36$ |

# A APPENDIX

## A.1 FASTER LORA CONVERGENCE

Table 10 demonstrates that LoRA fine-tuning with $\mathcal{L}_{\mathrm{LLM-JEPA}}$ loss not only achieves substantially higher accuracy than using $\mathcal{L}_{\mathrm{LLM}}$ alone, but also converges more quickly. Notably, at a LoRA rank of 512, our method already reaches accuracy comparable to full fine-tuning, whereas LoRA with only $\mathcal{L}_{\mathrm{LLM}}$ still exhibits a clear performance gap.

Table 11: Pretraining + fine-tuning Llama-3.2-1B-Instruct accuracy on pretraining dataset cestwc/paraphrase and fine-tuning dataset Rotten Tomatoes and Yelp by Next Token Prediction ($\mathcal{L}_{\mathrm{LLM}}$) loss vs. $\mathcal{L}_{\mathrm{LLM-JEPA}}$ loss (our method). Note that $\mathcal{L}_{\mathrm{LLM-JEPA}}$ is applied only at pretraining. We tune $lr_{pre}$ and $lr_{ft}$ by $\mathcal{L}_{\mathrm{LLM}}$, and stick to them in LLM-JEPA pretraining. We run pretraining 5 times, and for each pretrained model, we run fine-tuning 5 times. Average accuracy and standard deviation are reported. We also report $p$-value of paired, single-tailed $t$-Test.

| FT Dataset | Method | Accuracy (%) ↑ | $p$-value ↓ | Config |
|---|---|---|---|---|
| Rotten Tomatoes | $\mathcal{L}_{\mathrm{LLM}}$ | $56.57 \pm 1.66$ | $7.38e-4$ | $lr_{pre} = 8e-5, lr_{ft} = 4e-5$ |
| | $\mathcal{L}_{\mathrm{LLM-JEPA}}$ (ours) | $57.76 \pm 1.33$ | | $\lambda = 0.5, k = 2$, same $lr_{pre}, lr_{ft}$ |
| Yelp | $\mathcal{L}_{\mathrm{LLM}}$ | $26.46 \pm 0.92$ | $1.00e-3$ | $lr_{pre} = 8e-5, lr_{ft} = 8e-5$ |
| | $\mathcal{L}_{\mathrm{LLM-JEPA}}$ (ours) | $27.15 \pm 0.93$ | | $\lambda = 0.5, k = 2$, same $lr_{pre}, lr_{ft}$ |

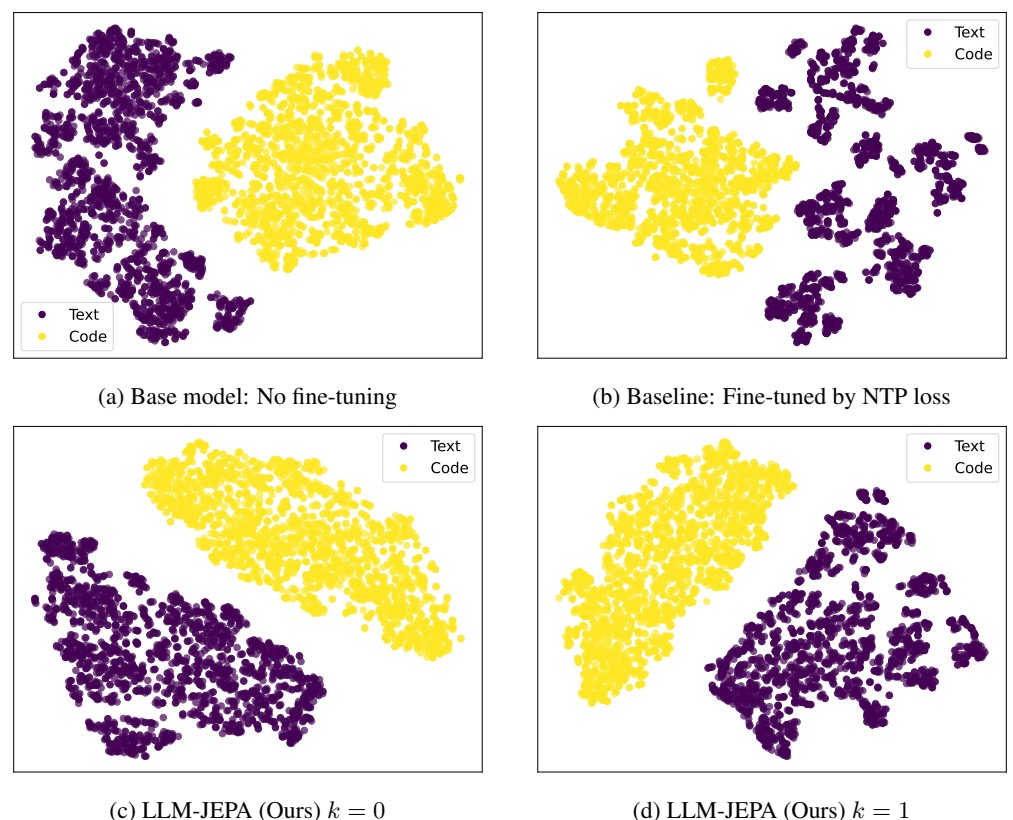

(a) Base model: No fine-tuning       (b) Baseline: Fine-tuned by NTP loss

(c) LLM-JEPA (Ours) $k = 0$      (d) LLM-JEPA (Ours) $k = 1$

Figure 6: $t$-SNE plot of f$Text$ and $Code$ representations in **(a)** Base mode without fine-tuning, **(b)** Baseline that is fine-tuned with NTP loss, **(c)** LLM-JEPA (ours) with $k = 0$, and **(d)** LLM-JEPA (ours) with $k = 1$. Clearly LLM-JEPA (ours) induced nice structure on the representations while fine-tuning with NTP loss disrupted the structure in the base model.

## A.2 LLM-JEPA INDUCES STRUCTURED REPRESENTATION

We present additional $t$-SNE plots of Text and Code representations in fig. 6, which show that different values of $k$ yield similar structural patterns. In contrast, standard fine-tuning appears to further disrupt the representation structure compared to the baseline model.

Table 12: Fine-tuning accuracy on dataset NL-RX-SYNTH with $\mathcal{L}_{\text{LLM}-\text{JEPA}}$ loss (ours) over various $\gamma/\lambda$. Configuration is $lr = 2e - 5, \lambda = 1, k = 0$. We maintain $\max(\gamma, \lambda) = 1.0$ to use a fixed $lr$. Each cell runs five times. Average accuracy and standard deviation are reported. When $\gamma = 0.0$, it generate only empty output.

| $\gamma/\lambda$ | Config | Accuracy (%) ↑ |
|---|---|---|
| 0.0 | $\gamma = 0.0, \lambda = 1.0$ | $0.00 \pm 0.00$ |
| 0.01 | $\gamma = 0.01, \lambda = 1.0$ | $1.38 \pm 0.06$ |
| 0.1 | $\gamma = 0.1, \lambda = 1.0$ | $45.80 \pm 5.04$ |
| 1.0 | $\gamma = 1.0, \lambda = 1.0$ | $70.42 \pm 2.36$ |
| 10.0 | $\gamma = 1.0, \lambda = 0.1$ | $67.52 \pm 1.45$ |
| 100.0 | $\gamma = 1.0, \lambda = 0.01$ | $66.83 \pm 3.89$ |
| $\infty$ | $\gamma = 1.0, \lambda = 0.0$ | $57.29 \pm 5.32$ |

| | $k=0$ | $k=1$ | $k=2$ | $k=3$ | $k=4$ |
|---|---|---|---|---|---|
| $\lambda = 0.5$ | 35.68% | 36.12% | 36.32% | 36.13% | 36.36% |
| $\lambda = 1.0$ | 36.03% | 33.83% | 33.01% | 33.86% | 36.15% |
| $\lambda = 2.0$ | 31.11% | 33.54% | 24.64% | 17.27% | 34.13% |

(a) Llama on GSM8K, $lr = 2e - 5$

| | $k=0$ | $k=1$ | $k=2$ | $k=3$ | $k=4$ |
|---|---|---|---|---|---|
| $\lambda = 0.5$ | 49.43% | 49.52% | 49.55% | 49.61% | 49.18% |
| $\lambda = 1.0$ | 48.13% | 48.54% | 49.00% | 50.55% | 50.13% |
| $\lambda = 2.0$ | 46.95% | 47.20% | 48.79% | 47.41% | 48.50% |

(b) Llama on Spider, $lr = 1e - 5$

| | $k=0$ | $k=1$ | $k=2$ | $k=3$ | $k=4$ |
|---|---|---|---|---|---|
| $\lambda = 0.5$ | 34.87% | 34.87% | 36.10% | 38.57% | 35.25% |
| $\lambda = 1.0$ | 34.87% | 35.60% | 36.32% | 37.28% | 38.36% |
| $\lambda = 2.0$ | 32.69% | 34.49% | 37.50% | 41.70% | 43.12% |

(c) Gemma on SYNTH, $lr = 1e - 5$

| | $k=0$ | $k=1$ | $k=2$ | $k=3$ | $k=4$ |
|---|---|---|---|---|---|
| $\lambda = 1.0$ | 14.18% | 10.52% | 13.53% | 12.96% | 17.92% |
| $\lambda = 2.0$ | 12.87% | 12.00% | 19.70% | 19.33% | 13.72% |
| $\lambda = 4.0$ | 15.63% | 13.11% | 18.67% | 25.40% | 16.63% |

(d) OpenELM on SYNTH, $lr = 8e - 5$

| | $k=0$ | $k=1$ | $k=2$ | $k=3$ | $k=4$ |
|---|---|---|---|---|---|
| $\lambda = 1.0$ | 87.43% | 87.38% | 87.30% | 83.40% | 83.53% |
| $\lambda = 2.0$ | 87.52% | 87.14% | 87.33% | 82.64% | 78.17% |
| $\lambda = 4.0$ | 87.12% | 87.10% | 87.30% | 87.27% | 87.36% |

(e) OLMo on SYNTH, $lr = 8e - 5$

| | $k=0$ | $k=1$ | $k=2$ | $k=3$ | $k=4$ |
|---|---|---|---|---|---|
| $\lambda = 0.5$ | 57.69% | 57.62% | 60.26% | 60.15% | 60.47% |
| $\lambda = 1.0$ | 57.65% | 58.43% | 59.92% | 59.78% | 59.74% |
| $\lambda = 2.0$ | 57.48% | 56.64% | 60.03% | 60.59% | 60.14% |

(f) Llama on SYNTH, Pretrain, $lr = 8e - 5$

Figure 7: In general we didn't find any pattern on where the best accuracy could appear. It could be at either high-end or low-end of either $\lambda$ or $k$. Furthermore, there can be dips and spikes in random locations. Nonetheless, adjacent cells have close accuracy most of times, and sweeping $(k, \lambda) \in \{0, 1, 2, 3, 4\} \times \{0.5, 1, 2, 4\}$ normally yield satisfiable results. Each cell is an average of five runs, $epoch = 4$.

## A.3 ABLATION STUDY ON THE ROLE OF $\mathcal{L}_{\text{LLM}}$

One limitation of eq. (2) is that the contribution of $\mathcal{L}_{\text{LLM}}$ cannot be effectively reduced to 0. To address this, we introduce an additional hyperparameter $\gamma$ to explicitly control its relative strength:

$$\mathcal{L}_{\text{LLM}-\text{JEPA}} = \gamma \times \underbrace{\sum_{\ell=2}^{L} \mathcal{L}_{\text{LLM}}(\text{Text}_{1:\ell-1}, \text{Text}_\ell)}_{\text{generative capabilities}} + \lambda \times \underbrace{d(\text{Pred}(\text{Enc}(\text{Text})), \text{Enc}(\text{Code}))}_{\text{abstraction capabilities}}, \quad (3)$$

We vary the ratio $\gamma/\lambda$ within $[0, 1]$ while enforcing $\max(\gamma, \lambda) = 1$ to maintain a constant learning rate. Table 12 shows that $\mathcal{L}_{\text{LLM}}$ remains essential for generative performance: when $\gamma = 0$, the fine-tuned model produces only empty outputs. This indicates that the JEPA component primarily serves as a regularization term, complementing the generative loss.

## A.4 ADDITIONAL GENERATION EXAMPLES

Table 13 presents additional examples generated by fine-tuning Llama-3.2-1B-Instruct on the NL-RX-SYNTH dataset using $\mathcal{L}_{\text{LLM}}$ and $\mathcal{L}_{\text{LLM}-\text{JEPA}}$, respectively.

Table 13: More regular expressions generated by Llama-3.2-1B-Instruct after fine-tuning with $\mathcal{L}_{\text{LLM}}$ loss and $\mathcal{L}_{\text{LLM−JEPA}}$ loss (ours). Color code: wrong , extra , missing

| Ground Truth | $\mathcal{L}_{\text{LLM}}$ | $\mathcal{L}_{\text{LLM−JEPA}}$ (ours) |
|---|---|---|
| **lines ending with a vowel or starting with a character** | | |
| ([AEIOUaeiou].*[A-Za-z].*)+ | ([AEIOUaeiou].*[A-Za-z].*)+ | ([AEIOUaeiou].*[A-Za-z].*)+ |
| **ines containing either a lower-case letter, a vowel, or a letter** | | |
| ((.*)([AEIOUaeiou]))l((.)(.*)) | (.*) ( ([AEIOUaeiou]) l((.)(.*)) ) | (.*) ( ([AEIOUaeiou]) l((.)(.*)) ) |
| **lines starting with the string 'dog' before a vowel** | | |
| ((([A-Za-z])7,).*(dog).* | ((([A-Za-z])7,).*(dog).* .* | ((([A-Za-z])7,).*(dog).* |
| **lines not containing a letter and the string 'dog'** | | |
| ((((A-Z])+)l([a-z]))(.*) | ((((A-Z])+)l([a-z]))(.*) + | ((((A-Z])+)l([a-z]))(.*) |
| **lines with a character before a vowel and the string 'dog', zero or more times** | | |
| .*(.)&([0-9])&(dog).* | .*(.)&([0-9])&(dog).* .* | .*(.)&([0-9])&(dog).* .*.* |
| **lines with a vowel at least once before not a character** | | |
| ((([A-Za-z])+).*(~([0-9])).* | ((([A-Za-z])+).*(~([0-9])).* .* | ((([A-Za-z])+).*(~([0-9])).* |

## A.5 OVERFITTING BEHAVIOR IN LORA FINE-TUNING

We also conducted experiments to examine whether LoRA fine-tuning with $\mathcal{L}_{\text{LLM-JEPA}}$ exhibits similar resistance to overfitting. As shown in fig. 8, accuracy under $\mathcal{L}_{\text{LLM-JEPA}}$ generally continues to improve with additional epochs, whereas fine-tuning with $\mathcal{L}_{\text{LLM}}$ shows clear signs of overfitting. Notably, the standard deviation is much higher than in full fine-tuning, likely reflecting the lower capacity of LoRA fine-tuning. An interesting pattern emerges: for $\mathcal{L}_{\text{LLM-JEPA}}$, larger standard deviations often coincide with dips in accuracy, whereas for $\mathcal{L}_{\text{LLM}}$ they tend to accompany accuracy spikes. This suggests that such fluctuations may be unreliable indicators of generalization quality.

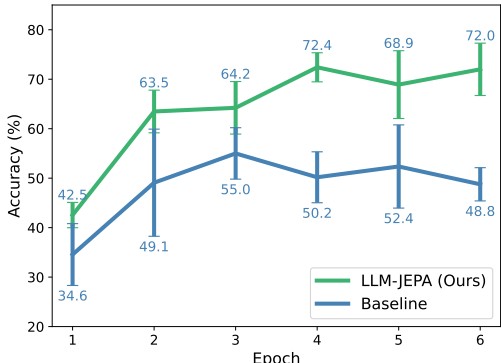

Figure 8: LLM-JEPA resists overfitting in LoRA fine-tuning. Fine-tuning with $\mathcal{L}_{\text{LLM−JEPA}}$ loss (our method) resists overfitting. When fine-tuning with $\mathcal{L}_{\text{LLM}}$ loss start to overfit, $\mathcal{L}_{\text{LLM−JEPA}}$ kept improving. However the trend is not as stable as in full fine-tuning, possibly due to limited capacity of LoRA fine-tuning.

## A.6 STRUCTURED REPRESENTATIONS INDUCED BY LLM-JEPA

We also examine the representation space to better understand how LLM-JEPA regularizes learned features. Specifically, we plot $t$-SNE embeddings for both Text and Code across three settings: the base model, a model fine-tuned with $\mathcal{L}_{\text{LLM}}$, and a model fine-tuned with $\mathcal{L}_{\text{LLM-JEPA}}$. As shown in fig. 4, clear structure emerges after fine-tuning with $\mathcal{L}_{\text{LLM-JEPA}}$. We hypothesize that $\mathcal{L}_{\text{LLM-JEPA}}$ enforces structure in the representation space by constraining the mapping from $\text{Enc}(\text{Text})$ to $\text{Enc}(\text{Code})$ within a narrow subspace. If this is the case, the SVD decomposition of $\text{Enc}(\text{Text}) − \text{Enc}(\text{Code})$ should yield significantly smaller singular values, which is confirmed in fig. 3. Furthermore, we hypothesize that the mapping is approximately linear. To test this, we compute the least-squares

Table 14: Fine-tuning accuracy on dataset NL-RX-SYNTH by Next Token Prediction ($\mathcal{L}_{\text{LLM}}$) loss vs. $\mathcal{L}_{\text{LLM-JEPA}}$ loss (our method). Each cell is the best possible accuracy over a set of configurations. Each configuration runs five times. Average accuracy and standard deviation are reported. We also report $p$-value of paired, single-tailed $t$-Test.

| Model | Method | Accuracy (%) ↑ | $p$-value ↓ | Config |
|---|---|---|---|---|
| Llama-3.2-1B-Instruct | $\mathcal{L}_{\text{LLM}}$ | $57.29 \pm 5.32$ | $1.0e-3$ | $lr = 2e-5$ |
| | $\mathcal{L}_{\text{LLM-JEPA}}$ (ours) | $71.46 \pm 1.34$ | | $\lambda = 1, k = 1$, same $lr$ |
| gemma-2-2b-it | $\mathcal{L}_{\text{LLM}}$ | $33.65 \pm 3.24$ | $5.5e-3$ | $lr = 1e-5$ |
| | $\mathcal{L}_{\text{LLM-JEPA}}$ (ours) | $43.12 \pm 2.61$ | | $\lambda = 2, k = 4$, same $lr$ |
| OpenELM-1_1B-Instruct | $\mathcal{L}_{\text{LLM}}$ | $12.07 \pm 1.81$ | $5.1e-4$ | $lr = 8e-5$ |
| | $\mathcal{L}_{\text{LLM-JEPA}}$ (ours) | $25.40 \pm 2.40$ | | $\lambda = 4, k = 3$, same $lr$ |
| OLMo-2-0425-1B-Instruct | $\mathcal{L}_{\text{LLM}}$ | $87.09 \pm 0.36$ | $2.5e-3$ | $lr = 8e-5$ |
| | $\mathcal{L}_{\text{LLM-JEPA}}$ (ours) | $87.52 \pm 0.29$ | | $\lambda = 2, k = 0$, same $lr$ |

Table 15: Fine-tuning accuracy by model Llama-3.2-1B-Instruct, $\mathcal{L}_{\text{LLM}}$ loss vs. $\mathcal{L}_{\text{LLM-JEPA}}$ loss (our method). Each cell is the best possible accuracy over a set of configurations. Each configuration runs five times. Average accuracy and standard deviation are reported. We also report $p$-value of paired, single-tailed $t$-Test.

| Dataset | Method | Accuracy (%) ↑ | $p$-value ↓ | Config |
|---|---|---|---|---|
| NL-RX-TURK | $\mathcal{L}_{\text{LLM}}$ | $22.49 \pm 1.91$ | $2.4e-4$ | $lr = 2e-5$ |
| | $\mathcal{L}_{\text{LLM-JEPA}}$ (ours) | $30.94 \pm 1.13$ | | $\lambda = 1, k = 1$, same $lr$ |
| GSM8K | $\mathcal{L}_{\text{LLM}}$ | $32.36 \pm 0.58$ | $9.6e-5$ | $lr = 2e-5$ |
| | $\mathcal{L}_{\text{LLM-JEPA}}$ (ours) | $36.36 \pm 0.20$ | | $\lambda = 0.5, k = 4$, same $lr$ |
| Spider | $\mathcal{L}_{\text{LLM}}$ | $47.52 \pm 2.44$ | $4.0e-3$ | $lr = 4e-5$ |
| | $\mathcal{L}_{\text{LLM-JEPA}}$ (ours) | $50.55 \pm 2.08$ | | $\lambda = 1, k = 3$, same $lr$ |

regression error, and table 16 supports this hypothesis. Together, these results suggest that LLM-JEPA promotes a near-linear transformation between Text and Code representations, which may underlie its accuracy improvements.

## A.7 PERFORMANCE ACROSS MODEL SIZES

We also evaluate LLM-JEPA across different model sizes. As shown in table 17, we observe statistically significant improvements at all scales. Since there is no official 8B version of Llama-3.2, we instead use Llama-3.1-8B-Instruct, where performance collapsed due to the model's difficulty in properly terminating regular expressions. To address this, we additionally evaluate using a `startswith` criterion—that is, a prediction is considered correct if the generated regular expression begins with the ground-truth expression, removing the need for exact termination. Under this metric, we again observe statistically significant accuracy improvements.

## A.8 EFFICIENT HYPERPARAMETER SEARCH

We found that the optimal $k$ values consistently cluster around 0–1 or 3–4, and moreover, when the optimum does not occur at $k = 0$ or $k = 3$, the optimal $\lambda$ is always the same $\lambda$ that performs best at those two anchor points. This yields an efficient search strategy:

Table 16: LLM-JEPA is almost a linear transformation from Enc(Text) to Enc(Code).

| | $\min_X \|\, \text{Enc(Text)} \cdot X - \text{Enc(Code)} \|_2$ | Avg. Top 100 Singular |
|---|---|---|
| Base model | 3953.11 | 310.73 |
| $\mathcal{L}_{\text{LLM}}$ | 3035.01 | 341.80 |
| LLM-JEPA (Ours) $k = 1$ | 4.47 | 94.84 |
| LLM-JEPA (Ours) $k = 0$ | 4.04 | 16.82 |

Table 17: Fine-tuning accuracy on NL-RX-SYNTH by Next Token Prediction ($\mathcal{L}_{\mathrm{LLM}}$) loss vs. $\mathcal{L}_{\mathrm{LLM-JEPA}}$ loss (our method). Each case runs five times. Average accuracy and standard deviation are reported. We also report $p$-value of paired, single-tailed $t$-Test. Note that Llama does not have official 3.2-8B, and we have to use 3.1-8B, which has a lower accuracy. Still LLM-JEPA sees significant improvement. We also evaluated on OLMo-2-7B.

| Model | Method | Accuracy (%) ↑ | $p$-value ↓ | Config |
|---|---|---|---|---|
| Llama-3.2-1B-Instruct | $\mathcal{L}_{\mathrm{LLM}}$ | $57.29 \pm 5.32$ | $1.0e-3$ | $lr = 2e-5$ |
| | $\mathcal{L}_{\mathrm{LLM-JEPA}}$ (ours) | $71.46 \pm 1.34$ | | $\lambda = 1, k = 1$, same $lr$ |
| Llama-3.2-3B-Instruct | $\mathcal{L}_{\mathrm{LLM}}$ | $74.55 \pm 3.58$ | $0.0352$ | $lr = 2e-5$ |
| | $\mathcal{L}_{\mathrm{LLM-JEPA}}$ (ours) | $77.16 \pm 3.66$ | | $\lambda = 2, k = 0$, same $lr$ |
| Llama-3.1-8B-Instruct | $\mathcal{L}_{\mathrm{LLM}}$ | $35.77 \pm 6.60$ | $0.0131$ | $lr = 2e-5$ |
| | $\mathcal{L}_{\mathrm{LLM-JEPA}}$ (ours) | $63.57 \pm 16.81$ | | $\lambda = 2.0, k = 0$, same $lr$ |
| OLMo-2-1124-7B-Instruct | $\mathcal{L}_{\mathrm{LLM}}$ | $87.26 \pm 0.27$ | $0.0345$ | $lr = 2e-5$ |
| | $\mathcal{L}_{\mathrm{LLM-JEPA}}$ (ours) | $87.75 \pm 0.33$ | | $\lambda = 20, k = 2$, same $lr$ |

|  | $k=0$ | $k=1$ | $k=2$ | $k=3$ | $k=4$ |
|---|---|---|---|---|---|
| $\lambda = 1/1024$ | 20.25% | 20.11% | 20.33% | 20.24% | 20.34% |
| $\lambda = 0.5$ | 19.90% | 19.64% | 19.73% | 20.01% | 20.06% |
| $\lambda = 1.0$ | 19.92% | 19.40% | 19.72% | 19.82% | 19.99% |
| $\lambda = 2.0$ | 20.06% | 20.21% | 19.86% | 19.75% | 19.67% |
| $\lambda = 4.0$ | 20.26% | 20.35% | 19.79% | 19.84% | 19.83% |
| $\lambda = 8.0$ | 20.12% | 20.55% | 20.00% | 20.34% | 20.12% |
| $\lambda = 16.0$ | 20.60% | 20.19% | 20.79% | 20.71% | 20.55% |
| $\lambda = 32.0$ | 20.18% | 20.93% | 20.14% | 20.29% | 20.50% |
| $\lambda = 64.0$ | 20.71% | 20.84% | 19.80% | 20.53% | 20.36% |
| $\lambda = 128.0$ | 20.79% | 21.04% | 20.37% | 20.84% | 20.61% |
| $\lambda = 1024.0$ | 21.59% | 21.29% | 20.99% | 21.42% | 20.46% |

Figure 9: Fine-tuning HellaSwag with Llama-3.2-1B allows $\lambda$ to be scaled up to 1024, with performance continuing to improve.

- Evaluate all $\lambda$ values at $k \in \{0, 3\}$ and identify the best $(\lambda, k)$.

- From the best $(\lambda, k)$, iteratively evaluate $(\lambda, k+1)$ until no further improvement is observed.

Under this scheme, the search cost is reduced from $N \cdot M$ to $2N + O(1)$, where $N$ is the number of $\lambda$ values and $M$ is the number of $k$ values. We verified that this procedure reliably recovers the optimal $(\lambda, k)$ across all experiments reported in the paper.

## A.9 EXTREMELY HIGH JEPA LOSS DROPOUT RATES

We conducted additional experiments to study the effect of JEPA loss dropout rates. At extremely high dropout $(1 - 0.0625)$, we observed that varying $\lambda$ does not yield meaningful accuracy improvements. This suggests a simplified search strategy: keep $\lambda$ fixed. Using this approach, we found that accuracy remains stable even at a dropout rate of $1 - 0.125$, better than previously identified $1 - 0.25$ (table 18).

Table 18: Varying $\lambda$ does not yield meaningful accuracy improvements at dropout rate $1 - 0.0625$.

| $1 - 0.5, \lambda = 1$ | $1 - 0.25, \lambda = 1$ | $1 - 0.125, \lambda = 1$ | $1 - 0.0625, \lambda = 1$ | $1 - 0.0625, \lambda = 2$ | $1 - 0.0625, \lambda = 0.5$ |
|---|---|---|---|---|---|
| $73.42 \pm 1.00$ | $73.31 \pm 0.76$ | $73.20 \pm 0.46$ | $71.53 \pm 1.27$ | $70.68 \pm 1.06$ | $70.63 \pm 1.17$ |

Table 19: Representation alignment leads to stronger extrapolation behavior.

| Config | LLM-JEPA | Base Model | Regular Fine-tuning |
|---|---|---|---|
| **Cosine similarity** | $0.995 \pm 0.002$ | $0.914$ | $0.698 \pm 0.049$ |
| **Accuracy** | $95.56 \pm 6.09$ | $0$ | $88.89 \pm 0.0$ |

### A.10 REPRESENTATION ALIGNMENT ANALYSIS

Figure 6 shows that LLM-JEPA preserves and even improves the alignment of text representations, whereas standard fine-tuning disrupts this alignment. Intuitively, a well-aligned representation space should facilitate **extrapolation** and improve **generalization**. To test this hypothesis, we constructed a controlled dataset that maps Text of the form `"lines with a number repeated k or more times"` to the corresponding Code (regular expression) `"([0-9])k,"` for $k \in [1, 9]$.

We measured the **minimum pairwise cosine similarity** among the representations. LLM-JEPA collapses these representations almost onto a single line (min cosine similarity $\approx 0.995$), outperforming both the base model and the regularly fine-tuned model. Moreover, the regularly fine-tuned model fails to generalize to the unseen case $k = 1$ (absent in the training data), whereas LLM-JEPA can generalize correctly. This supports the intuition that improved representation alignment leads to stronger extrapolation behavior (table 19). Note that `"1,"` is equivalent to `"+"`, but the regularly fine-tuned model failed both.

### A.11 $\ell_2$-NORM

The number of $\ell_2$-norm loss in table 3 is correct. Note that the magnitude of the $\ell_2$-norm loss is equivalent to the MSE loss multiplied by the embedding dimension $N$ and for morden LLMs, $N$ is very large. In table 3, MSE performs reasonably well, but when switching to $\ell_2$-norm the effective scale of the loss is amplified by a factor of $N$. This dramatically increases the magnitude of the JEPA term, which can destabilize optimization and lead to collapse. This explains the severe drop in accuracy when using $\ell_2$-norm.

