# OpenReview forum: "LLM-JEPA: Large Language Models Meet Joint Embedding Predictive Architectures"
_ICLR.cc/2026/Conference — ICLR 2026 Poster_

### Official Review · Reviewer_iw7d · 2025-10-17

**Soundness:** 3
**Presentation:** 3
**Contribution:** 3
**Rating:** 8
**Confidence:** 4

**Summary:**

This paper, LLM-JEPA, brings the Joint-Embedding Predictive Architecture (JEPA) that was originally introduced in the ViT domain to LLMs (specifically, text-code generation LLMs). In the vision domain, JEPA self-supervisedly to learn to predict the representations of some target image patches, given some context image patches. The underlying assumption is that all patches represent the same thing. In this paper, LLM-JEPA self-supervisedly learn to predict the representations of the code, given prompt text, and the underlying assumption is that the prompt and the code represent the same thing. LLM-JEPA is done simply by adding additional predictor tokens and a prediction loss. The authors show that this method improves standard LLM metrics significantly without overfitting.

**Strengths:**

-	Although the JEPA idea is not new, bringing it to the LLM domain is novel.
-	The authors acknowledge the underlying assumption of JEPA: they require multiple views of the *same* information as in text-code generation, but also go further to explore how it generalizes to QA tasks.
-	The authors applied Loss Dropout and observed dual wins in saving compute and improving accuracy.
-	The writing flows naturally (especially with the help of the natural questions), and is a pleasure to read. Having pseudo code, implementation (eg, Line 152) is a strong plus.

**Weaknesses:**

-	Line 161: recommend adding a simple diagram to illustrate how the Pred() is added to an LLM transformer. The description is a bit confusing: beyond the standard transformer, is there a separate prediction network? Are the predictors also generated in k auto-regressive loops or one-shot?
-	Recommend showing a visualization of the mask in Line 188. Readers can imagine this looks like two upper-triangular matrices along the diagonal, but adding a diagram makes it more intuitive.
-	Table 1 has no description in the main text. Does it intend to show failure cases?
-	(Minor) The citation style does not follow the ICLR 2026 template https://iclr.cc/Conferences/2026/AuthorGuide (i.e., xxx et al.).

**Questions:**

-	In Eq.1, recommend noting somewhere what XEnt() stands for. Typically, people just write CrossEntropy. Similarly, does NTP stand for next token prediction? Please note it somewhere.
-	In Line 161, is the [PRED] added after the concatenated text-code sequence or in between? (as mentioned above, a diagram would simplify things here).
-	In Line 166, could the authors elaborate on what it means to use the embedding of the last predictor token? Is it the hidden_state?
-	Line 170 typo: “lies in obtained” -> “lies in obtaining”?
-	In Line 190, could the authors explain which two forward passes? What are the purposes of each two?
-	In Figure 3, the blue curve is when k=0, but in Line 166, the authors say in this case the predictor is trivial. Does being trivial mean JEPA is turned off, or mean it is most effective?
-	In Figure 3, right side, we observe that even the NTP finetuning loss is almost the same at stabilization (yellow/blue), but at evaluation time their accuracy in fact differs a lot (51 vs 71). This is interesting. Do the authors have a hypothesis for the reason?
-	Line 292: Are the wrong/missing/extra labels compared against the GT or any valid regex? The proposed method’s output at line 292 has two additional `{}`.
-	Line 373: using L2 norm degrades accuracy from 71% to 2%, which is surprising how sensitive it is to the distance metric. Is this number correct?

---

> ### Author Response · Authors · 2025-11-17
>
> _Thank you so much for catching many subtle but important issues. All of them will be fixed in the next revision._
>
> > is there a separate prediction network? Are the predictors also generated in k auto-regressive loops or one-shot?
>
> In our method, there is no separate prediction network. We append $k$ [PRED] tokens to the end of the `Text`, run the entire sequence through the LLM, and use the final-layer hidden state of the last [PRED] token as the predicted representation. Functionally, this can be viewed as the model learning to generate [PRED_1] conditioned on the `Text`, then autoregressively generating [PRED_2] conditioned on `Text` $\oplus$ [PRED_1], and so on through [PRED_k]. See Figure 1 in https://bit.ly/llm-jepa-additional-charts (shared anonymously) for an illustration.
>
> That said, we conducted additional experiments with a standalone linear predictor, which **underperforms** compared to predictor tokens. See **Standalone Linear Predictors** in the response to all reviewers.
>
> > Recommend showing a visualization of the mask in Line 188.
>
> The mask is an additive attention mask consisting of two lower-triangular regions filled with 0s and all other positions set to $−\infty$. This structure preserves causal attention within each view while blocking cross-view attention. See Figure 2 in https://bit.ly/llm-jepa-additional-charts (shared anonymously) for a visualization.
>
> > Table 1 has no description in the main text. Does it intend to show failure cases?
>
> Thank you for catching these subtle but important issues. We will correct them in the next revision. Table 1 is intended to illustrate all four combinations of success and failure for LLM and LLM-JEPA. Line 292 is meant to show a case where both methods succeed, and each cell should contain `{}`. Because `{}` is invisible in LaTeX, we manually prepend a backslash (`\`), and we inadvertently omitted this formatting for the first two columns. This will also be fixed in the next revision.
>
> > (Minor) citation style.
>
> Thank you for catching these subtle but important issues. We will correct them in the next revision.
>
> > In Eq.1, XEnt() and NTP.
>
> Yes, XEnt refers to **CrossEntropy**, and NTP stands for **next-token prediction**. We will clarify both in the next revision.
>
> > In Line 161, is the [PRED] added after the concatenated text-code sequence or in between?
>
> We append them to the end of `Text`. See Figure 1 in https://bit.ly/llm-jepa-additional-charts (shared anonymously) for an illustration.
>
> > In Line 166, Is it the hidden_state?
>
> Yes, it’s the `hidden_state`. See Figure 1 in https://bit.ly/llm-jepa-additional-charts (shared anonymously) for an illustration.
>
> > Line 170 typo: “lies in obtained” -> “lies in obtaining”?
>
> Yes, will fix in the next revision
>
> > In Line 190, two forward passes?
>
> The LLM fine-tuning step already performs one forward pass. We require one additional pass to obtain the encodings of `Text` and `Code`. We will update the wording to “one extra forward pass”.
>
> > In Figure 3, predictor is trivial.
>
> In mathematics, a “trivial function” usually refers to an identity function. We misused the term in the paper. In the case $k = 0$, the "predictor simply becomes an identity mapping", not that JEPA is turned off. We will revise the text accordingly.
>
> > In Figure 3, NTP finetuning loss is almost the same
>
> A similar phenomenon has been observed in prior work (e.g., https://arxiv.org/pdf/2402.11337, see Figures 5 and 6). Although that study focuses on vision models, it shows that reconstruction loss—analogous to NTP in our setting—can be nearly identical across different models, yet their downstream performance can differ substantially. The key insight is that reconstruction error does not uniquely determine a model’s function in representation space. Two models can achieve the same reconstruction loss while learning qualitatively different internal representations, leading to divergent downstream accuracies.
>
> We believe a similar effect is occurring here.
>
> > Line 292: GT or any valid regex? two additional `{}`.
>
> The comparison is made against the ground truth. In Line 292, all columns are supposed to contain `{}`. Because `{}` is invisible in LaTeX, we prepend a backslash (`\`) to make it visible, and we inadvertently omitted this formatting for the first two columns. We will correct this in the next revision.
>
> > Line 373: L2 norm degrades accuracy from 71% to 2%
>
> Yes, the number is correct. Note that the magnitude of the L2 loss is equivalent to the MSE loss multiplied by the embedding dimension $N$ and for morden LLMs, $N$ is very large. In Table 3, MSE performs reasonably well, but when switching to L2 the effective scale of the loss is amplified by a factor of $N$. This dramatically increases the magnitude of the JEPA term, which can destabilize optimization and lead to collapse. This explains the severe drop in accuracy when using L2.

---

> ### Author Response · Authors · 2025-11-24
>
> Dear Reviewer iw7d,
>
> Just a gentle note to say we would be glad to answer any additional questions you might have. Please feel free to reach out if there is anything we can further clarify during the discussion period. Thank you again for your thoughtful review.

---

> ### Comment · Reviewer_iw7d · 2025-11-25
>
> Thanks to the authors for making the clarifications.  My concerns are addressed.

---

> > ### Author Response · Authors · 2025-11-25
> >
> > Thank you so much for your thoughtful engagement and for letting us know that your concerns have been addressed — we truly appreciate it.
> >
> > I just wanted to kindly check whether you had intended to update your score, or if you feel that the current score already reflects your assessment. No pressure at all either way — we completely respect your judgment.
> >
> > Thank you again for the time and care you’ve put into reviewing our work.

---

### Official Review · Reviewer_waki · 2025-10-27

**Soundness:** 3
**Presentation:** 3
**Contribution:** 3
**Rating:** 6
**Confidence:** 3

**Summary:**

This paper proposes LLM-JEPA, a novel training objective that brings Joint Embedding Predictive Architectures (JEPA) into large language models. By combining the standard autoregressive next-token prediction loss with a JEPA-style embedding prediction loss, the model improves representation quality while maintaining generative capability.

**Strengths:**

1. The work systematically introduces JEPA into LLM training for the first time, bridging a long-standing gap between vision and language representation learning.

2. Demonstrates consistent empirical improvements across diverse architectures, scales, and datasets.

3. Provides clear analysis of embedding structure (t-SNE, SVD), showing how JEPA regularizes the representation space.

4. Maintains generation ability while enhancing abstraction and generalization.

**Weaknesses:**

1. Training cost is substantially higher due to multiple forward passes (≈2× compute), as the authors acknowledge.

2. Hyperparameter tuning (λ, k) appears expensive and unstable across settings.

3. The approach depends on multi-view data (e.g., text–code pairs), limiting applicability to generic text-only corpora.

4. The paper could discuss more about how JEPA objectives affect downstream reasoning or interpretability beyond accuracy metrics.

5. It would be interesting to see whether JEPA acts more like regularization or representation alignment—the current discussion is mostly empirical.

**Questions:**

1. Could JEPA be extended to pure text tasks using data augmentation (e.g., paraphrase or Q–A pairs)?

2. How sensitive is the method to the choice of distance metric (cosine vs. L2)?

3. During large-scale pretraining, how do you plan to manage compute cost — would loss dropout scale effectively to billions of tokens?

4. Did you observe any degradation in generation diversity or fluency due to the embedding-space constraint?

5. Can the JEPA loss be interpreted as encouraging semantic disentanglement or latent structure emergence?

---

> ### Author Response · Authors · 2025-11-17
>
> > Training cost is substantially higher due to multiple forward passes.
>
> We agree that the additional training cost is a significant obstacle for broad adoption, and reducing this overhead is a top priority for our future work. That said, as the reviewer also noted, JEPA loss dropout provides an effective mitigation mechanism.
>
> > Hyperparameter tuning $(\lambda, k)$ appears expensive and unstable across settings.
>
> Thank you for pointing this out. We examined this issue carefully and identified an empirical structure that dramatically reduces the search cost from $N\cdot M$ to $2N + O(1)$, where $N$ is the number of $\lambda$ values and $M$ is the number of $k$ values. We verified that this procedure reliably recovers the optimal $(\lambda, k)$ across all experiments reported in the paper. See **Efficient Hyperparameter Search** in the response to all reviewers.
>
> > The approach depends on multi-view data (e.g., text–code pairs), limiting applicability to generic text-only corpora.
>
> We agree that reliance on a two-view structure is the main current limitation, and we aim to relax or eliminate this requirement in our future work.
>
> We also want to clarify that, although the current formulation requires two views, the method already applies more broadly in practice: beyond naturally paired domains such as text–code, we have successfully extended it to GSM8K (figure 1), NQ-Open, and HellaSwag (table 4), none of which have a clear two-view structure.
>
> > The paper could discuss more about how JEPA objectives affect downstream reasoning or interpretability beyond accuracy metrics.
>
> We agree that understanding how JEPA-style objectives influence downstream reasoning and interpretability is an important direction. We plan to investigate these aspects in future work, once we have addressed the two primary limitations—training cost and reliance on a two-view structure.
>
> > It would be interesting to see whether JEPA acts more like regularization or representation alignment—the current discussion is mostly empirical.
>
> We conducted additional analyses (see **Representation Alignment Analysis** in the response to all reviewers), which indicate that the JEPA loss behaves more like a **representation alignment** mechanism than a generic regularizer. Specifically, it encourages representations of semantically similar texts to collapse onto a straight line, which in turn facilitates **extrapolation** and improves **generalization**.
>
> > Could JEPA be extended to pure text tasks using data augmentation (e.g., paraphrase or Q–A pairs)?
>
> Yes—data augmentation is a promising path toward removing the reliance on a two-view structure. In fact, we have already taken preliminary steps in this direction: we pretrained on the `cestwc/paraphrase` dataset (Table 9), which led to strong downstream fine-tuning performance.
>
> > How sensitive is the method to the choice of distance metric (cosine vs. L2)?
>
> As shown in Table 3, alternative distance metrics (L2, MSE, etc.) consistently underperform cosine similarity. Intuitively, cosine similarity encourages representations to align along a one-dimensional direction, whereas L2-based metrics do not enforce such directional alignment. This is consistent with our analysis in **Representation Alignment Analysis**, which suggests that the performance gains are primarily driven by improved **representation alignment**.
>
> > During large-scale pretraining, how do you plan to manage compute cost — would loss dropout scale effectively to billions of tokens?
>
> As noted earlier, reducing compute overhead is a top priority for our future work, especially for large-scale pretraining. We have conducted preliminary experiments on the `cestwc/paraphrase` dataset (Table 9), which contains **11 million tokens**, and observed that the performance trends remain consistent even at this larger scale. Although this is still far from the scale of modern foundation models, the stability of the results is encouraging and suggests that JEPA loss dropout may continue to be effective when scaling to much larger corpora.
>
> > Did you observe any degradation in generation diversity or fluency due to the embedding-space constraint?
>
> We evaluated generation quality by producing HellaSwag sentence-completion samples. We did not observe any noticeable differences in diversity or fluency. Sample outputs are available https://bit.ly/llm-jepa-samples (shared anonymously).
>
> > Can the JEPA loss be interpreted as encouraging semantic disentanglement or latent structure emergence?
>
> In LLM-JEPA, JEPA loss primarily encourages **representation alignment**, which is conceptually closer to the emergence of a latent structure rather than semantic disentanglement. Please see **Representation Alignment Analysis** in the response to all reviewers for details. Additionally, as shown in Table 3, InfoNCE performs substantially worse, suggesting that a contrastive objective—typically more aligned with disentanglement—can be detrimental in this setting.

---

> ### Author Response · Authors · 2025-11-24
>
> Dear Reviewer waki,
>
> We wanted to kindly check whether you have any remaining questions or points you would like us to address. We are happy to clarify anything further. Thank you again for your time and feedback.

---

> ### Comment · Reviewer_waki · 2025-11-26
>
> Thank you for the rebuttal. The clarifications were helpful and addressed some earlier concerns, particularly regarding stability and distance metrics. Some broader questions such as the mechanism behind representation alignment and the dependence on two-view data remain open. So I am keeping my original score.

---

> > ### Author Response · Authors · 2025-11-26
> >
> > > Some broader questions such as the mechanism behind representation alignment and the dependence on two-view data remain open
> >
> > Thank you for clearly articulating your remaining concerns. We revisited your original comments and realized that our rebuttal may have missed addressing a few key points. We would be grateful for the opportunity to clarify them here.
> >
> > **For Representation Alignment Analysis**
> >
> > We realized that our rebuttal may have missed a few key links in the full argument. The complete analysis has **four parts**, but in our earlier response we inadvertently omitted the first and last components. For completeness, we summarize the full picture below:
> > 1. **Prior work indicates that SGD acts as an implicit regularizer** [1], and that deep neural networks naturally confine learned representations to a low-dimensional manifold [2]. To illustrate this phenomenon in our setting, we constructed a controlled dataset mapping phrases of the form `lines with a number repeated k or more times` to the corresponding regular expression `([0-9]){k,}` for $k \in [1, 9]$. For the base model (`Llama-3.2-1B-Instruct`), the minimum cosine similarity among representations of the `Text` (NL descriptions) is **0.914**, while that of the `Code` (regular expressions) is **0.805**.
> > 2. **Despite both similarities being high, the `Code` representations are noticeably less aligned.** This may be because regular expressions are relatively rare in pretraining data. In the paper, we show that LLM-JEPA can align the `Code` representation toward the `Text` representation (Table 14, Fig. 3 left). By aligning `Code` to `Text`, LLM-JEPA increases the minimum cosine similarity of `Code`. Because alignment is bi-directional, this creates a positive feedback loop. As a result, both `Text` and `Code` representations in our controlled dataset collapse onto an almost straight line (minimum cosine similarity at **0.995**). Intuitively, extrapolation is much easier along a one-dimensional manifold. Indeed, the LLM-JEPA–fine-tuned model successfully **generalizes to the unseen case** $k =1$, which does not appear in training.
> > 3. **Regular fine-tuning disrupts both `Text` and `Code` manifolds** (minimum cosine similarity at **~0.7**), and the regularly fine-tuned model fails to generalize to the unseen $k=1$ case. This contrast supports the hypothesis that representation alignment is a key driver behind LLM-JEPA’s superior extrapolation ability.
> > 4. **Importantly, LLM-JEPA does not collapse all regular expressions and NL descriptions into a single line**. On the entire SYNTH test dataset, the minimum cosine similarity is **0.619 ± 0.056**, showing that alignment is structured and selective—not indiscriminate collapse.
> >
> > ... on Controlled Dataset | Base model | Regular Fine-tuning | LLM-JEPA
> > -|-|-|-
> > Min Cosine Similarity of `Code` | 0.805 | 0.714 $\pm$ 0.028 | 0.995 $\pm$ 0.002
> > Min Cosine Similarity of `Text` | 0.914 | 0.698 $\pm$ 0.049 | 0.995 $\pm$ 0.002
> > NL $\rightarrow$ Regex Accuracy | 0 | 88.89 $\pm$ 0.0 | 95.56 $\pm$ 6.09
> >
> > In summary, LLM-JEPA improves extrapolation and generalization by aligning the underlying representations of `Text` and `Code`.
> >
> > **For Dependency on Two-View Data**
> >
> > We also realized that our rebuttal did not fully emphasize a point already discussed in the paper (Lines 417–423): we have **already extended our method to QA (NQ-Open) and sentence completion (HellaSwag)**—both of which lack an explicit two-view structure.
> >
> > For **QA**, we treat the question as `Text` and the answer as `Code`. Although there is no explicit pairing in the dataset, pairs such as “The capital of France” and “Paris” clearly refer to the same underlying concept. In this sense, QA data often contains an **implicit conceptual two-view structure**, even if not explicitly labeled as such.
> >
> > For **sentence completion**, we use the leading context as `Text` and the completion sentence as `Code`. The structure here is less obvious, but the context still provides meaningful predictive information about the completion. While this setting requires additional scaffolding, it again demonstrates that our method can operate without an explicit two-view design.
> >
> > Overall, both experiments show that **our approach does not rely on an explicit two-view structure**, and can naturally extend to tasks where the views are implicit or inferred.
> >
> > We hope this additional explanation provides helpful context, and we sincerely appreciate your careful consideration of our work. If there are any remaining questions, we would be more than happy to address them.
> >
> > **Reference**
> >
> > [1] Implicit Regularization in Matrix Factorization, NeurIPS 2017, https://arxiv.org/abs/1705.09280
> >
> > [2] Intrinsic dimension of data representations in deep neural networks, NeurIPS 2019 https://arxiv.org/abs/1905.12784

---

### Official Review · Reviewer_qoq5 · 2025-10-30

**Soundness:** 3
**Presentation:** 3
**Contribution:** 3
**Rating:** 4
**Confidence:** 3

**Summary:**

This paper proposes LLM-JEPA, a method that augments standard autoregressive LLM training (next-token prediction) with a JEPA-style embedding-space objective that forces embeddings of two views to align via a predictor network. The JEPA term is added to the generative cross-entropy loss with a balancing weight λ, and the predictor is implemented by appending small numbers of special [PRED] tokens that reuse model weights. The authors implement a custom attention mask to obtain separate view embeddings efficiently, introduce random JEPA-loss dropout to reduce compute overhead, and provide a thorough empirical study across multiple model families (Llama3, Gemma2, OpenELM, OLMo). They show consistent accuracy gains, representation improvements (t-SNE, SVD, near-linear mapping from text to code embeddings), and robustness to overfitting; they also provide ablations on loss choices, predictor placement, and loss dropout.

**Strengths:**

1. The formulation (LLLM + λ · JEPA) and practical predictor design (tied weights via [PRED] tokens) are simple, elegant, and directly applicable to existing transformer models without architectural surgery.
2. Results include full fine-tuning and LoRA, pretraining experiments, several datasets (including code and QA tasks), multiple model sizes, statistical significance testing (five seeds), and useful visualizations (t-SNE, SVD) that support the claimed representational benefits.

**Weaknesses:**

1. Most experiments are fine-tuning or pretraining on relatively small/targeted corpora. While dropout amortizes cost, it remains unclear how LLM-JEPA behaves (performance, wall-clock, memory) at large pretraining scales used for modern foundation models.
2. The paper presents empirical evidence (SVD, near-linear mapping), but lacks deeper theoretical analysis or causal ablations that would clarify when and why the embedding alignment improves downstream generation and reasoning

**Questions:**

1. The paper reports grid search results and suggests keeping λ·(1−α) constant with dropout, but could you provide concrete heuristics or automated tuning strategies? How sensitive are gains to λ when moving to very different view types or sequence lengths?
2. The method uses the last-token hidden state as the embedding and appends [PRED] tokens as a tied predictor. Have you tried averaging representations, CLS-like tokens, multilayer predictors, or decoupled predictors? How robust are the results to these choices?

---

> ### Author Response · Authors · 2025-11-17
>
> > While dropout amortizes cost, it remains unclear how LLM-JEPA behaves (performance, wall-clock, memory) at large pretraining scales used for modern foundation models.
>
> Thank you for highlighting this limitation. We agree that understanding how LLM-JEPA behaves at large pretraining scales is essential for broad adoption, and this is the top priority of our future work.
>
> That said, we have already provided preliminary pretraining results. In particular, we experimented with the `cestwc/paraphrase` dataset (Table 9), which contains **11 million tokens**, and observed that the same performance trends hold in this larger-scale setting. While this is far from the scale of modern foundation models, the consistency of the results is encouraging and suggests that the method may transfer to larger pretraining regimes.
>
> > The paper … lacks deeper theoretical analysis or causal ablations that would clarify when and why the embedding alignment improves downstream generation and reasoning
>
> We hypothesize that the downstream improvements arise from **representation alignment**—that is, the model learns to collapse representations of semantically similar text into a tightly aligned region (often nearly a line), which in turn enables better **extrapolation** and strengthens **generalization**. We validated this hypothesis through additional experiments; please see **Representation Alignment Analysis** in the response to all reviewers.
>
> Furthermore, Table 3 shows that InfoNCE performs substantially worse, which is consistent with our hypothesis: its contrastive objective explicitly pushes representations apart, counteracting the alignment behavior that appears critical for extrapolative reasoning.
>
> > The paper reports grid search results and suggests keeping $\lambda\cdot (1−\alpha)$ constant with dropout, but could you provide concrete heuristics or automated tuning strategies? How sensitive are gains to $\lambda$ when moving to very different view types or sequence lengths?
>
> Thank you for raising this point. We conducted additional experiments to better understand the interaction between $\lambda$ and the loss dropout rate $\alpha$. We found that when $\alpha$ is pushed to extremely high dropout (i.e., an $1−\alpha$ close to 1), varying $\lambda$  no longer yields meaningful gains. This observation suggests a simplified tuning heuristic: **keep $\lambda$  fixed** when searching over $\alpha$.
>
> Using this strategy, we were able to push the dropout rate to $1−\alpha=0.125$ with stable accuracy. At $1−\alpha=0.0625$, we begin to observe accuracy degradation. Please see **Exploring JEPA Loss Dropout Rates** in the response to all reviewers for detailed results.
>
> > The method uses the last-token hidden state as the embedding and appends [PRED] tokens as a tied predictor. Have you tried averaging representations, CLS-like tokens, multilayer predictors, or decoupled predictors? How robust are the results to these choices?
>
> We conducted additional experiments with two alternatives—averaging representations and using a standalone linear predictor network—and both underperform LLM-JEPA. Please see **Averaging Representations** and **Standalone Linear Predictors** in the response to all reviewers. Intuitively, in decoder-only LLMs the final token already attends to all preceding tokens, so its hidden state subsumes their information; averaging all token representations tends to blur this signal. For the linear predictor, we believe the main issue is the **cold-start** problem: the predictor must be trained entirely from scratch, making optimization noticeably more brittle.
>
> We did not experiment with a `CLS`-like token, as this mechanism is uncommon in decoder-only architectures, which form the backbone of nearly all modern LLMs. We also did not explore multilayer predictor networks; as shown in Table 14 and Fig. 3 (left), the mapping from `Text` to `Code` representations is nearly linear, suggesting that a linear predictor should already be sufficient.

---

> ### Author Response · Authors · 2025-11-24
>
> Dear Reviewer qoq5,
>
> Thank you for your thoughtful reassessment and for updating your score. We sincerely appreciate your careful reading and engagement with the paper. If you have any remaining questions or would like clarification on any aspect, we would be very glad to address them.
>
> Thank you again for your time.

---

> > ### Comment · Reviewer_qoq5 · 2025-11-27
> >
> > Thanks for making the clarifications. My concerns have been addressed and I raised my score to 6.

---

### Official Review · Reviewer_j7Db · 2025-10-31

**Soundness:** 3
**Presentation:** 2
**Contribution:** 3
**Rating:** 6
**Confidence:** 3

**Summary:**

This paper proposes a novel training objective $\mathcal{L}_{\mathrm{LLM-JEPA}}$ for LLM training, aiming at introducing the successful JEPA insights, i.e., directly predicting from embedding space to absorb abstraction knowledge, from vision into the LLM studies, which mainly rely on the reconstruction ability on token level. LLM-JEPA recognizes the naturally paired data (e.g., text & code or Q & A in this paper) as different views of the same thing. The objective is to predict from the embeddings of one view to the embeddings of another view. Experiments show the performances of LLM-JEPA on both the reconstruction ability and predictability.

**Strengths:**

- The idea of introducing JEPA from vision to language is novel. Compared to the representation learning in LLM, which mainly focuses on obtaining good embeddings, LLM-JEPA doesn’t neglect its generative capabilities, and experiments show that the additional JPEA loss helps both pretraining and finetuning.
- I like the analysis in the experiment part, including showing that next-token prediction loss cannot optimize the JPEA objective and visualizing the t-SNE and the singular value to verify the strengths of LLM-JEPA.

**Weaknesses:**

- Though efficient, I doubt the usage of this method, especially as its strong dependency on multi-view data, which may limit the application to real-world industrial LLMs. Do you have some blueprints of the future work for more universal scenarios (e.g., unsupervised pretraining via data augmentation on texts that don’t have paired views)?
- Another weakness, based on my understanding, is the additional training complexity and hyperparameter search cost. From Fig. 7 in appendix, there is no clear law of the best $(\lambda, k)$ combination, which is also mentioned in the limitation part.

**Questions:**

- For the predictor, you append $k$ `[PRED]` tokens to reuse the model’s weights. Can you elaborate more on the design intuition of this setting, especially when $k>0$? Why sometimes a larger $k$ would be better?
- In Table 3, I find that InfoNCE surprisingly performs much worse than metrics. Do you have some explanations on this? Does it just mean the predictability is more crucial than the contrastiveness in this task?
- In this paper, both the text and code encoders are initiated with a same LLM with same weights. In prior practice in JPEA [1], the target encoder is an EMA of the context encoder weights, which can enhance the training stability. Can the EMA design be applied to LLM-JPEA?

## References

[1] Self-Supervised Learning from Images with a Joint-Embedding Predictive Architecture. CVPR 2023.

---

> ### Author Response · Authors · 2025-11-17
>
> > Do you have some blueprints of the future work for more universal scenarios (e.g., unsupervised pretraining via data augmentation on texts that don’t have paired views)?
>
> Thank you for the question. We agree that reliance on a two-view structure is the main current limitation, and we aim to relax or eliminate this requirement in our future work. Our ongoing roadmap—still preliminary but active—is to:
> 1. Remove the need for explicit paired views, potentially via data augmentation or alternative mechanisms for constructing compatible predictive targets.
> 2. Eliminate the extra training cost so the method is compute-neutral relative to standard pretraining and/or fine-tuning.
> 3. Develop the approach into a general paradigm for universal pretraining.
>
> We also want to clarify that, although the current formulation requires two views, the method already applies more broadly in practice: beyond naturally paired domains such as text–code, we have successfully extended it to GSM8K (figure 1), NQ-Open, and HellaSwag (table 4), none of which have a clear two-view structure. This demonstrates that the technique is already useful in a wide range of settings, even before the full removal of the two-view constraint.
>
> > Another weakness, based on my understanding, is the additional training complexity and hyperparameter search cost.
>
> Thank you for pointing this out. We agree that additional hyperparameter search could be a concern, so we examined this issue carefully and identified an empirical structure that dramatically reduces the search cost from $N\cdot M$ to $2N + O(1)$, where $N$ is the number of $\lambda$ values and $M$ is the number of $k$ values. We verified that this procedure reliably recovers the optimal $(\lambda, k)$ across all experiments reported in the paper. See **Efficient Hyperparameter Search** in the response to all reviewers.
>
> > Can you elaborate more on the design intuition of this setting, especially when $k > 0$? Why sometimes a larger $k$ would be better?
>
> Our intuition follows from how LLMs are pretrained: they are optimized to predict the next token. In our setting, we ask the model to predict a code representation from a text representation. Introducing a special [PRED] token leverages the pretrained next-token prediction mechanism for this new predictive task, making it a natural extension of the existing architecture.
>
> For $k > 0$, we use multiple predictor tokens: [PRED_1], [PRED_2], …, [PRED_k], and take the hidden state of [PRED_k] as the predicted representation. The intuition is straightforward: more predictor tokens allocate more FLOPs to the prediction pathway, which can increase prediction precision. This is conceptually similar to the use of more `<think>` tokens in reasoning models.
>
> We also performed additional experiments to determine whether performance gains are driven primarily by the **number** of predictor tokens or the **variety** of their embeddings, and to evaluate whether predictor tokens can be replaced by a standalone linear predictor network. Please see **Predictor Token Design** and **Standalone Linear Predictors** in the response to all reviewers.
>
> > I find that InfoNCE surprisingly performs much worse than metrics. Do you have some explanations on this?
>
> We hypothesize that the accuracy improvements stem from **representation alignment**—i.e., the model learns to collapse representations of semantically similar text into a tightly aligned region (often nearly a line), which facilitates **extrapolation** and improves **generalization**. Under this view, InfoNCE can be detrimental because its contrastive objective explicitly pushes representations apart, counteracting the alignment needed for extrapolative behavior.
>
> We validated this hypothesis through additional experiments; please see **Representation Alignment Analysis** in the response to all reviewers.
>
> > In this paper, both the text and code encoders are initiated with a same LLM with same weights. In prior practice in JPEA [1], the target encoder is an EMA of the context encoder weights, which can enhance the training stability. Can the EMA design be applied to LLM-JPEA?
>
> In principle, an EMA target encoder could be added to LLM-JEPA, but we deliberately avoid it for several reasons:
> 1. EMA was originally introduced in BYOL to prevent representational collapse when no auxiliary objective is present. In LLM-JEPA, the NTP loss is always preserved, which guarantees non-collapse.
> 2. Maintaining an EMA encoder requires keeping a full second copy of the model. For large LLMs, this is prohibitively memory-intensive, and we aim to keep the method as efficient as possible.
> 3. A separate EMA encoder prevents us from computing representations in a single forward pass. This would increase the overhead from approximately 2$\times$ to 3$\times$ (without dropout mitigation), working directly against our goal of reducing training cost.

---

> > ### Comment · Reviewer_j7Db · 2025-11-19
> >
> > Thanks for your detailed response! The response addresses my concerns in the initial review. The attempt to enhance LLM with JEPA representation ability in this paper is persuasive. I learned a lot from this paper. One key insight delivered by this paper is that, rather than focusing only on reconstruction, enriching its representation is essential and beneficial, and does not significantly harm the model (the authors' response to Q4 of the reviewer qoq5). The better generalization might result from the reduced complexity from a well-structured embedding space to the final objective compared to a model w/o such constraints.
> >
> > The discussion of future work is also promising. Overall, I think this paper is of high quality. After carefully reading other reviewers' comments and the authors' responses, I have no more concerns. Thus, I will raise my score from 6 to 8.

---

> > > ### Author Response · Authors · 2025-11-19
> > >
> > > Thank you so much for the thoughtful update and for taking the time to engage deeply with our work. Much appreciated!

---

### Author Response · Authors · 2025-11-17
**Additional Experiments and Analyses**

_Thank you for the constructive feedback from all reviewers. We conducted a series of additional experiments in response to the comments, and we summarize the key findings below._

## Efficient Hyperparameter Search

We found that the optimal $k$ values consistently cluster around 0–1 or 3–4, and moreover, when the optimum does not occur at $k = 0$ or $k = 3$, the optimal $\lambda$ is always the same $\lambda$ that performs best at those two anchor points. This yields an efficient search strategy:

1. Evaluate all $\lambda$ values at $k \in$ {0, 3} and identify the best $(\lambda, k)$.
2. From the best $(\lambda, k)$, iteratively evaluate $(\lambda, k + 1)$ until no further improvement is observed.

Under this scheme, the search cost is reduced from $N\cdot M$ to $2N + O(1)$, where $N$ is the number of $\lambda$ values and $M$ is the number of $k$ values. We verified that this procedure reliably recovers the optimal $(\lambda, k)$ across all experiments reported in the paper.

## Predictor Token Design

For $k > 0$, we append multiple predictor tokens: [PRED_1], …, [PRED_k], to the text and use the hidden state of [PRED_k] as the predicted representation. To better understand the mechanism, we examined whether performance gains come primarily from (i) the **number** of predictor tokens or (ii) the **variety** of distinct predictor tokens. We conducted ablations using identical predictor tokens, and the results (see table below) show only minor differences. This suggests that the dominant factor is simply increasing the number of prediction steps (FLOPs), rather than requiring distinct token embeddings.

Config | Distinct PRED | Same PRED
-|-|-
Llama 3.2 / GSM8K | 36.36 $\pm$ 0.20 | 36.74 $\pm$ 0.70
OpenELM / SYNTH | 25.40 $\pm$ 2.40 | 25.01 $\pm$ 1.60

## Standalone Linear Predictors

An alternative design is to replace the [PRED] tokens with a standalone linear predictor network. A linear predictor is sufficient because the mapping from `Text` to `Code` representations is nearly linear (Table 14, Fig. 3 left). However, in the fine-tuning regime this approach suffers from a **cold-start problem**—the linear head must be trained entirely from scratch. Empirically, a straightforward linear predictor yields consistently subpar performance compared to using [PRED] tokens (last column).

$\lambda=0.5$, linear | $\lambda=1$, linear | $\lambda=2$, linear | $\lambda=1, k=1$
-|-|-|-
70.16$\pm$1.87 | 67.93$\pm$2.70 | 64.54$\pm$4.99 | 71.46$\pm$1.34

## Representation Alignment Analysis

Figure 6 shows that LLM-JEPA preserves and even improves the alignment of text representations, whereas standard fine-tuning disrupts this alignment. Intuitively, a well-aligned representation space should facilitate **extrapolation** and improve **generalization**. To test this hypothesis, we constructed a controlled dataset that maps `Text` of the form `“lines with a number repeated k or more times”` to the corresponding `Code` (regular expression) `“([0-9]){k,}”` for $k \in [1, 9]$.

We measured the **minimum pairwise cosine similarity** among the representations. LLM-JEPA collapses these representations almost onto a single line (min cosine similarity $\approx$ 0.995), outperforming both the base model and the regularly fine-tuned model. Moreover, the regularly fine-tuned model fails to generalize to the unseen case $k = 1$ (absent in the training data), whereas LLM-JEPA can generalize correctly. This supports the intuition that improved representation alignment leads to stronger extrapolation behavior. Note that `{1,}` is equivalent to `+`, but the regularly fine-tuned model failed both.

Config | LLM-JEPA | Base Model | Regular Fine-tuning
-|-|-|-
Cosine similarity | 0.995$\pm$0.002 | 0.914 | 0.698$\pm$0.049
Accuracy | 95.56$\pm$6.09 | 0 | 88.89$\pm$0.0

## Exploring JEPA Loss Dropout Rates

We conducted additional experiments to study the effect of JEPA loss dropout rates. At extremely high dropout (1 – 0.0625), we observed that varying $\lambda$ does not yield meaningful accuracy improvements. This suggests a simplified search strategy: **keep $\lambda$ fixed**. Using this approach, we found that accuracy remains stable even at a dropout rate of 1 – 0.125, better than previously identified 1 – 0.25.

1 - 0.5, $\lambda=1$ | 1 - 0.25, $\lambda=1$ | 1 - 0.125, $\lambda=1$ | 1 - 0.0625, $\lambda=1$ | 1 - 0.0625, $\lambda=2$ | 1 - 0.0625, $\lambda=0.5$
-|-|-|-|-|-
73.42$\pm$1.00 | 73.31$\pm$0.76 | 73.20$\pm$0.46 | 71.53$\pm$1.27 | 70.68$\pm$1.06 | 70.63$\pm$1.17

## Averaging Representations

We performed an additional ablation in which we replaced the final hidden state (hidden\_state[-1]) with the average of all hidden states. This modification degraded performance.

Averaging Representation | hidden\_state[-1]
-|-
65.46$\pm$3.51 | 71.46$\pm$1.34

---

### Author Response · Authors · 2025-11-28
**To the new AC**

To the new AC,

We are very sorry for the situation surrounding the incident. While we fully respect the ICLR Program Committee’s decision to revert all score changes as a necessary measure to preserve scientific integrity, we would like to share a few pieces of context for your consideration:
* Reviewers j7Db and qoq5 updated their scores (to 8 and 6, respectively) **on November 19th, well before the incident was reported**. The timestamps of their review modifications can be seen in the screenshot at: https://bit.ly/llm-jepa-review-screenshot
 (shared anonymously).
* Reviewer waki (original score 6) noted that several concerns had been addressed but still had open questions regarding representation alignment and the dependence on two-view data. In our last comment, we provided detailed clarifications addressing both points, though we did not have the opportunity to hear their follow-up response.
* Reviewer iw7d (original score 8) indicated that all concerns were resolved but did not update the score. In our final message, we gently asked whether they wished to revise their assessment, but again, we did not have the chance to receive a reply.

Taken together, all reviewers expressed positive sentiment toward the contribution—either _before_ the discussion opened or on dates clearly _predating_ the reported incident. The discussion was moving in a constructive and **upward trajectory**, and it is possible that additional updates might have occurred had the process not been interrupted.

We understand that this situation has created extra work for the ACs, and we are sincerely grateful for your time and care in handling a complex issue. We very much appreciate your willingness to review all materials with fresh eyes and trust your judgment in making a fair and thoughtful recommendation.

Thank you again for the additional effort and for helping uphold the high standards that make ICLR such a strong and principled community.

---

### Meta-Review · Area_Chair_nRmz · 2026-01-02

**Summary:**

The authors made a successful rebuttal that the score raised from 4668 to 6688. Based on the discussion, most concerns are solved by the rebuttal. After reading the paper and discussion, the AC tends to accept this paper.

**Reviewer Concerns:**

According to the discussion, Reviewer waki think "some broader questions such as the mechanism behind representation alignment and the dependence on two-view data remain open"

**Reviewer Scores:**

According to the discussion, all the reviewers have made their final decision.

---

### Decision · Program_Chairs · 2026-01-26

Accept (Poster)